# Reprogramming Hairy Root Cultures: A Synthetic Biology Framework for Precision Metabolite Biosynthesis

**DOI:** 10.3390/plants14131928

**Published:** 2025-06-23

**Authors:** Chang Liu, Naveed Ahmad, Ye Tao, Hamad Hussain, Yue Chang, Abdul Wakeel Umar, Xiuming Liu

**Affiliations:** 1College of Life Sciences, Engineering Research Center of the Chinese Ministry of Education for Bioreactor and Pharmaceutical Development, Jilin Agricultural University, Changchun 130118, China; lc2518441830@163.com (C.L.); 15584405809@163.com (Y.T.); cy18391348615@163.com (Y.C.); 2Institute for Safflower Industry Research of Shihezi University/Pharmacy College of Shihezi University/Key Laboratory of Xinjiang Phytomedicine Resource and Utilization, Ministry of Education, Ministry of Education, Shihezi 832003, China; naveed@sjtu.edu.cn; 3Department of Agriculture, Faculty of Chemical and Life Sciences, Abdul Wali Khan University Mardan, Mardan 23200, Pakistan; hammadagarian629@gmail.com; 4BNU-HKUST Laboratory of Green Innovation, Advanced Institute of Natural Sciences, Beijing Normal University at Zhuhai (BNUZ), Zhuhai City 519087, China

**Keywords:** hairy roots, *Agrobacterium rhizogenes*, bioreactor, specialized metabolite, crop resilience, sustainable development

## Abstract

Hairy root cultures induced by *Agrobacterium rhizogenes* (*Rhizobium rhizogenes*) provide a sustainable approach to meet the growing demand for economically valuable plant-derived compounds in the face of depleting natural resources. These cultures exhibit rapid, hormone-independent growth and genetic stability, making them viable for producing bioactive compounds, plant-specialized metabolites, and recombinant proteins. However, challenges remain in optimizing large-scale production, improving bioreactor efficiency, and enhancing metabolite synthesis across different plant species. This review addresses these challenges by exploring the mechanisms behind the induction of hairy root cultures, their applications in genetic and metabolic engineering, and their potential in environmental remediation. The review further highlights recent advances in biotechnology and illustrates how the hairy root system can sustainably meet industrial, pharmaceutical, and agricultural needs. In addition, by pointing out essential research areas such as optimizing culture conditions, increasing metabolite yields, and scaling up production, this work strengthens the significance of hairy root cultures in meeting the demand for high-value products while ensuring sustainable resource utilization. In particular, the integration of hairy root systems with advanced genomic tools such as transcriptomics and CRISPR technology holds immense potential for accelerating pathway-specific metabolic engineering, enhancing biosynthetic flux, and expanding their applications in sustainable agriculture and pharmaceutical innovation. This convergence is expected to drive substantial economic value by optimizing the production of high-value bioactive compounds, improving crop resilience, and facilitating precision medicine. Future work involving systems and synthetic biology will be instrumental in unlocking novel functions and ensuring broader deployment of hairy root cultures across industrial biotechnological platforms.

## 1. Introduction

The utilization of biological resources is essential for the production of high-value bioactive compounds. With the rapid pace of economic development and technological innovation, the demand for medicinal, food, energy, and industrial products is increasing, leading to a notable strain on natural biological resources [1,2]. This pressing issue has elevated concerns regarding the sustainable management and recycling of these resources, particularly as traditional extraction techniques often fall short in meeting contemporary demands [3]. In recent years, hairy root cultures have emerged as a viable solution in this context, providing a platform that offers a straightforward induction process, rapid growth, and genetic stability without the need for exogenous plant hormones, thereby facilitating the production of valuable bioactive compounds including alkaloids, phenolic compounds, flavonoids, and various other biologically active substances [4,5]. Their ability to provide a stable and high-yielding system for metabolite production makes them a promising tool for both pharmaceutical and industrial applications. Recent studies highlight the growing potential of hairy root cultures as an alternative platform for recombinant protein production, with ongoing improvements in culture conditions, genetic engineering techniques, and bioprocess optimization.

Hairy roots were first proposed and developed by Chilton in the 1980s [6]. They are highly differentiated adventitious roots produced by the injured parts of plants after infection with *Agrobacterium rhizogenes* (*A. rhizogenes)* [7]. Recent advancements in bioreactor technology have enabled the scalable and efficient production of bioactive compounds through hairy root cultures. For instance, hairy roots have been widely applied for decades in plant regeneration [8], production of specialized metabolites [9], varietal improvement [10], environmental remediation [11], biotransformation [12], and production of recombinant proteins [13]. These systems not only improve biomass yield but also enhance the stability and consistency of specialized metabolite production. The application of techniques such as temporary immersion systems can further augment this process by optimizing nutrient uptake and gaseous exchange [14]. Additionally, the integration of nanoparticle technology in plant tissue culture has demonstrated positive effects on the biosynthesis of phytochemicals, suggesting that combining hairy root cultures with innovative elicitors could significantly enrich the yield of bioactive compounds [15,16]. This development aligns with contemporary approaches toward green extraction methods that emphasize sustainability and eco-friendliness, addressing both productivity and environmental concerns in the process [3].

Medicinal plants have long served as essential sources of bioactive compounds, particularly specialized metabolites, which exhibit a wide range of pharmacological activities including anti-inflammatory, antimicrobial, anticancer, and antioxidant properties [17]. These metabolites, often produced in response to environmental stimuli, play critical roles in both traditional and modern medicine, contributing to the development of drugs like aspirin (derived from salicylic acid in willow bark) and paclitaxel (sourced from the Pacific yew tree) [18,19]. However, the extraction and purification of these natural products are often hindered by low yields, low purity, complex processing methods, high costs, and prolonged cultivation periods, threatening biodiversity and ecosystem sustainability. [20]. These limitations not only complicate research efforts but also contribute to resource depletion due to the overexploitation of medicinal plant species, threatening biodiversity and ecosystem sustainability. The harvesting of plants like ginseng and goldenseal has led to their near-endangerment in certain regions. To address these limitations, researchers from around the world have focused on developing alternative strategies for the efficient synthesis and production of active ingredients and specialized metabolites at reduced costs.

Although the hairy root culture system offers substantial advantages as a biotechnological platform, several critical challenges must be addressed to fully harness its potential. The long-term genetic stability of hairy root cultures and their ability to maintain consistent bioactive compound profiles remain insufficiently understood. In this review, we shed light on interdisciplinary approaches and technological innovations required for transforming hairy root cultures into a sustainable and scalable solution for pharmaceutical, agricultural, and environmental applications [21]. The possible use of hairy root systems across four primary domains, including target product biosynthesis, transgenic plant research, environmental remediation, and interfacing with other platforms, is discussed (Figure 1). While several review papers have explored various aspects of hairy root cultures, such as induction mechanisms, metabolic engineering, and their biotechnological applications [7,22,23,24], few have provided a comprehensive synthesis of recent innovations involving cutting-edge tools like CRISPR/Cas-based genome editing, artificial intelligence-driven optimization, and next-generation bioreactor systems. This review uniquely focuses on these advanced strategies to not only improve metabolite biosynthesis but also address long-standing challenges in scalability and consistency. Fostering a deeper commitment to these areas can significantly advance the future of hairy root culture, paving the way for the sustainable, scalable production of high-value bioactive compounds that meet global demands.

## 2. Induction Mechanism of Hairy Roots Cultures via T-DNA Insertion and Rol Genes Cluster

The induction of hairy root cultures is facilitated by the integration of the T-DNA (transferred DNA) region from the root-inducing (Ri) plasmid of *A. rhizogenes* into the plant genome [8]. *A. rhizogenes* harbors a Ri plasmid that contains specific T-DNA segments essential for autonomous hairy root formation, differing from the crown gall-inducing *A. tumefaciens*. This transformation process relies on the activation of the *virulence* (*Vir*) genes, which facilitate T-DNA excision and transfer from the bacterial plasmid into the plant genome via a type IV secretion system (T4SS) [25]. Upon integration, genes within the T-DNA region, particularly *root locus* (*rol*) genes, modulate phytohormonal signaling and auxin homeostasis, leading to the characteristic high-proliferation root phenotype [26].

The transformation is initiated when *A. rhizogenes* detects wounded plant tissue, triggering *vir* gene activation in response to phenolic compounds like acetosyringone. The VirD1/VirD2 complex processes the T-DNA into a single strand, and VirE2 proteins shield it, facilitating transfer to the plant nucleus [27]. This complex is transferred into the plant cell via T4SS, transported to the nucleus, and integrated into the host genome through illegitimate recombination, ultimately leading to the formation of transgenic hairy roots [25]. The T-DNA region contains key genes essential for hairy root formation, including *rolA*, *rolB*, *rolC*, and *rolD* (Figure 2), which regulate plant hormonal balance by modulating auxin and cytokinin homeostasis [28]. For example, rolA influences cytokinin metabolism and cell differentiation; rolB enhances auxin sensitivity; rolC lowers cytokinin levels to promote elongation; and rolD affects sugar metabolism [29]. Additionally, *aux1* and *aux2* genes encode enzymes involved in auxin biosynthesis, increasing indole-3-acetic acid (IAA) levels to further enhance root development [30]. Transformation efficiency is influenced by bacterial strains (A4, LBA9402, K599), plant species, co-cultivation parameters (e.g., acetosyringone level, duration), wounding methods, and selection strategies (e.g., cefotaxime and marker genes) [31].

Agrobacterium-mediated transformation remains a widely utilized genetic engineering tool due to its high efficiency, relatively low cost, and minimal genomic rearrangements compared to direct DNA delivery techniques like biolistics [32]. This system is particularly valuable for species recalcitrant to conventional tissue culture regeneration methods. Recent technological improvements have incorporated CRISPR/Cas-based genome editing, enabling precise functional genomics and pathway engineering in hairy root systems. Case studies include CRISPR-edited hairy roots in cucumber for gene knockout [10,33], in peanut for nodulation pathway analysis [34], and in *Brassica napus* for functional gene editing [35], demonstrating its broad applicability across species. The ability of hairy roots to stably express transgenes while maintaining biosynthetic capabilities makes them a powerful tool for metabolic engineering, production of pharmaceutical compounds, and synthetic biology applications.

## 3. Major Factors Affecting Hairy Root Induction for the High-Yield Production of Specialized Metabolites

Dicotyledonous plants induce hairy roots more effectively than monocotyledonous plants because of their cellular structure and phenolic compound release [36]. Hairy root induction has been successfully achieved in over 400 plant species, showcasing the broad host range of *A. rhizogenes*, particularly within dicotyledonous plants and, to a lesser extent, certain monocotyledonous species [37]. However, the production of specialized metabolites in hairy root cultures is strongly influenced by various biological, genetic, and environmental factors associated with induction protocols. Factors influencing the induction rate of hairy roots for improved biosynthesis of bioactive compounds, such as specialized metabolites, are complex and multifactorial, involving plant species, explant type, bacterial strain, infection method, and culture conditions.

### 3.1. Plant Species and Explant Type

The choice of plant species and the type of explants used for hairy root induction have profound effects on both the transformation success rate and the yield of specialized metabolites. Dicotyledonous plants are more responsive to transformation due to their compatible cellular architecture and higher levels of phenolic compounds, which act as chemical signals activating the virulence genes of *A. rhizogenes* [38]. These compounds facilitate the reprogramming of plant cells by *A. rhizogenes*, which is essential for root formation and subsequent metabolite production. The explant type (e.g., cotyledons, hypocotyls, roots, leaves) significantly affects both transformation efficiency and the resulting metabolite profiles. For instance, the selection of cotyledons, hypocotyls, and roots as explants can impact both the induction efficiency and metabolite composition [39]. A study on *Akebia trifoliata* revealed that variations in explant type, co-cultivation duration, and Agrobacterium strain significantly influenced the flavonoid content in hairy root cultures [40]. Similarly, using root explants in *Codonopsis pilosula* enhanced induction frequency and improved alkaloid and flavonoid yields. In *Linum mucronatum*, hypocotyl explants led to high production of valuable lignans such as podophyllotoxin and aryltetralin derivatives, which are important anticancer metabolites [41].

### 3.2. Agrobacterium Strains and Infection Methods

The strain of *A. rhizogenes* used is a critical factor influencing both root induction efficiency and specialized metabolite production when developing the hairy root culture system. First and foremost, the efficiency of transformation is highly dependent on the specificity, virulence, and plasmid type of the *A. rhizogenes* strain. Well-documented strains such as A4, A7, ArQual, K599, C58C1, MSU440, ATCC15834, LBA9402, R1000, R1200, pRi8196, pRiTR7, pRi1724, and A13 exhibit diverse levels of transformation efficiency and specialized metabolite induction capacity across various host plants [42]. For example, when the ArQual strain was used with an optimal bacterial concentration (OD600 0.5–0.6) for co-culture in *Camellia sinensis*, a significant accumulation of catechins was observed, emphasizing the importance of selecting both the strain and its co-cultivation parameters [43]. Similarly, a study on *Ocimum basilicum* compared several *A. rhizogenes* strains, including ATCC 13333, ATCC 15834, A4, R1000, R1200, and R1601. Their findings showed that R1601 strain achieved the highest transformation efficiency (94%), resulting in robust hairy root growth and elevated levels of rosmarinic acid (a valuable secondary metabolite) [44]. The use of R1601, C58C1, and A4 strains in *Physalis peruviana* was effective in inducing hairy roots, with strain A4 yielding the highest transformation frequency (72.31%) [45]. Notably, R1601-induced hairy roots exhibited enhanced antioxidant activity, correlating with elevated phenolic compound production. Furthermore, in Betula pendula, strain LBA9402 demonstrated the highest infection rate (92.7%) and promoted considerable biosynthesis of betulin and betulinic acid, two triterpenoids with pharmaceutical value [46]. R1601, LBA9402, and R1000 were also evaluated in *Morus alba*, where significant differences in transformation efficiency and metabolite accumulation were reported.

The infection methodology is another critical determinant of transformation success and metabolite output. Techniques like soaking, injection, or encapsulation can affect how well *Agrobacterium* establishes itself within plant cells. For instance, encapsulation of *A. rhizogenes* using alginate beads significantly improved root induction and metabolite stability in *Glycine max* (soybean) by promoting consistent bacterial–plant interactions [47]. In contrast, in *Corydalis yanhusuo*, explant soaking yielded higher transformation frequencies and increased tetrandrine content compared to the injection method [48]. Conversely, in *Capsicum annuum* (chili pepper), the injection method facilitated superior transformation frequency and enhanced capsaicinoid production, a trait of commercial interest in pharmaceuticals and food technology [49]. Taken together, the optimization of both bacterial strain selection and infection technique, tailored to specific host species and desired metabolites, is essential for achieving high-yield hairy root systems suitable for industrial applications.

### 3.3. Pre-Culture and Co-Culture Conditions

The pre-culture and co-culture conditions are critical determinants of both root induction and specialized metabolite production in hairy root cultures [42,50]. The pre-culture period serves as a crucial recovery phase for the explants, allowing them to heal from injury and begin cell division, which is necessary for the successful integration of T-DNA from *A. rhizogenes* [42]. Similarly, the co-culture period plays a vital role in ensuring efficient T-DNA transfer and subsequent root induction [51,52]. Research has demonstrated that extending the co-culture period within certain limits can enhance transformation efficiency and the production of specialized metabolites. For example, in *Echinacea purpurea*, a longer co-culture time led to increased levels of caffeic acid derivatives and other phenolic compounds [53,54]. On the other hand, prolonged co-cultivation beyond optimal limits can lead to negative effects, such as the overgrowth of *Agrobacterium*, which can result in explant browning [55,56]. This browning is typically a sign of microbial overgrowth, which can reduce explant viability and hinder root induction. As a consequence, metabolite yields may decrease due to impaired plant cell metabolism and excessive bacterial activity. In *Withania somnifera*, a 3-day co-culture period was found optimal for withanolide production, whereas longer durations led to tissue necrosis and lower yields [57]. Similarly, in *Hyoscyamus muticus*, transformation efficiency and tropane alkaloid accumulation were highest after a 48 h co-culture period [58]. In the case of *Artemisia annua*, a 2-day co-culture enhanced artemisinin biosynthesis while avoiding oxidative damage from prolonged bacterial interaction [59]. These findings across various plant systems underscore the critical need for species-specific optimization of co-culture duration. Prior studies on *C. annuum* also showed that a co-culture period of more than 7 days led to reduced capsaicinoid production, potentially due to excessive Agrobacterium proliferation [60,61]. However, the extended co-culture periods (e.g., >3 days in *Paeonia lactiflora*) led to bacterial overgrowth, causing explant browning and tissue necrosis due to oxidative stress and nutrient competition [62]. Thus, carefully optimizing the pre-culture and co-culture periods is crucial for maximizing both transformation efficiency and specialized metabolite accumulation in plant tissue cultures.

### 3.4. Growth Regulators and Medium Composition

The type and concentration of growth regulators in the culture medium are critical factors influencing both hairy root induction and specialized metabolite production. Auxins, such as indole-3-acetic acid (IAA) and naphthalene acetic acid (NAA), as well as cytokinins such as benzylaminopurine (BAP) and kinetin, are commonly used to promote root growth and enhance the establishment of hairy roots [63,64,65]. These growth regulators support root induction and stimulate the production of high-value bioactive compounds produced by plants. For instance, in *Origanum vulgare* (oregano), the optimization of medium composition led to an 81.5% induction frequency of hairy roots, accompanied by high yields of essential oils, highlighting the importance of growth regulators in metabolite production [66]. Furthermore, elicitor treatment with methyl jasmonate and ammonium nitrate enhanced thymol production in hairy root cultures of *Satureja sahendica*, suggesting a strategy to enhance specialized metabolite yields in this medicinal plant [67]. Similarly, in *Salvia miltiorrhiza*, methyl jasmonate and yeast extract were used to enhance tanshinone and phenolic acid content [68]. In *Glycyrrhiza uralensis*, chitosan and salicylic acid significantly improved glycyrrhizin production, demonstrating that elicitor selection and timing are key [69].

Moreover, the composition of the medium—especially nutrient balance, pH, and elicitor supplementation (e.g., acetosyringone, methyl jasmonate, chitosan, salicylic acid)— has been demonstrated to directly influence both transformation efficiency and specialized metabolite accumulation [4,70,71]. The consideration of medium composition also affects the growth environment of the plant cells, likely influencing metabolite production. For example, in *Codonopsis pilosula*, the correct combination of growth regulators and medium components resulted in enhanced production of saponins [72], demonstrating that both the physical and biochemical conditions of the culture medium must be carefully tailored for optimal metabolite biosynthesis. Moreover, the modification of medium pH and nutrient composition in hairy root culturing has been shown to further enhance metabolite yields [23,24]. Therefore, systematic optimization of growth regulators, medium composition, and elicitors is essential for achieving high transformation efficiencies and optimizing the production of valuable, specialized metabolites in hairy root cultures.

### 3.5. Environmental Factors

Environmental factors, including light, temperature, and humidity, play a significant role in the growth and development of hairy roots, as well as in the production of specialized metabolites [73,74]. Among these, temperature and light have been particularly studied in hairy root systems for their role in metabolic reprogramming. For example, in *S. miltiorrhiza*, cultivating hairy roots at 25 °C significantly increased tanshinone production, while temperatures above 30 °C reduced both biomass and metabolite content [75]. However, deviations from optimal temperature or fluctuating light conditions can disrupt metabolic pathways, potentially leading to either the enhancement or suppression of specific specialized metabolites. For instance, temperature stress in hairy root culture can trigger plant stress responses, often leading to the overproduction of certain bioactive compounds as a defense mechanism [76]. Conversely, extreme temperature fluctuations may cause metabolic imbalances that inhibit metabolite production [76]. Similarly, in a study involving *Hypericum perforatum*, hairy roots exposed to low-light conditions demonstrated enhanced biomass accumulation, whereas high-intensity light triggered the biosynthesis of hypericin and other naphthodianthrones [77]. On the other hand, prolonged exposure to light can sometimes induce specialized metabolite biosynthesis in certain plant species, as light acts as an environmental cue for specialized metabolic pathways [78].

The successful optimization of induction conditions for hairy root cultures requires a comprehensive understanding of both biological and environmental factors. In particular, determining optimal light cycles, temperature ranges, and humidity conditions tailored for each species is essential. For example, *Atropa belladonna* hairy roots required a 16:8 h light:dark cycle to increase tropane alkaloid production, highlighting how environmental parameters must be fine-tuned per species. By carefully manipulating these variables, it is possible to significantly enhance the production of bioactive compounds. Table 1 presents a summary of recent advancements in optimizing these conditions and their direct impact on metabolite yields. A deeper understanding of these crucial factors will continue to enhance the applications of hairy root cultures across various biotechnological sectors.

## 4. Metabolic Reprogramming in Hairy Root Cultures for High-Yield Specialized Metabolite Production

Hairy root cultures, induced by *A. rhizogenes*-mediated transformation, have become a cornerstone in synthetic biology for their unparalleled ability to produce, modify, and scale up plant-derived specialized metabolites. Recent advances in genetic engineering, biosynthetic plasticity, elicitation strategies, and bioreactor technologies have further enhanced their potential, allowing for the precise modulation of biosynthetic pathways and large-scale metabolite production. This section explores key breakthroughs in hairy root-based metabolic engineering, innovative elicitation strategies, scalable bioreactor approaches, and emerging technologies, offering a comprehensive insight into their role in biotechnology and pharmaceutical industries.

### 4.1. Metabolic Engineering for Pathways Optimization

Hairy root cultures offer a powerful platform for the metabolic engineering of plant-specialized metabolite biosynthesis. Through targeted genetic modifications, researchers have successfully redirected metabolic flux, overexpressed key transcription factors, and introduced heterologous genes to enhance the production of high-value phytochemicals [86,87,88]. For instance, CRISPR-Cas9 genome editing has been instrumental in optimizing biosynthetic pathways, as demonstrated in *Artemisia annua*, where knockout of *squalene synthase* (*SQS*) redirected precursor flux toward artemisinin biosynthesis in hairy roots, resulting in a 3.2-fold increase in yield [89,90]. Similarly, overexpression of *SmMYB1* in *S. miltiorrhiza* hairy roots activated genes in the diterpenoid biosynthetic pathway [91], leading to an 8.5-fold enhancement in tanshinone production. Metabolic pathway rewiring through combinatorial overexpression strategies has also been proven effective in improving metabolite yields. For example, the co-overexpression of squalene epoxidase and *cytochrome P450* genes in *W. somnifera* significantly increased withanolide accumulation, highlighting the potential of multigene engineering in hairy root systems [92]. Moreover, the *Rosmarinic acid synthase (RAS)* and *CYP98A14* genes from *S. miltiorrhiza* have been successfully introduced into the hairy root cultures [93]. The resulting transgenic lines overexpressing either *RAS* or *CYP98A14* exhibited a remarkable increase in phenolic acid production, with accumulation levels reaching up to three-fold higher than non-transgenic lines [93]. Overexpression of the *chalcone synthase* (*CHS*) gene from *G. uralensis* in its hairy root cultures showed a significant increase in total flavonoid content and the accumulation of liquiritigenin, isoliquiritigenin, and isoliquiritin [94]. Genetically modified hairy root cultures of *Valeriana officinalis*, overexpressing *farnesyl pyrophosphate synthase, valerendiene synthase, or germacrene C synthase* genes, exhibited a 2- to 4-fold increase in valerenic acid, a 1.5- to 4-fold rise in sesquiterpene hydrocarbons, and a 5- to 9-fold increase in oxygenated sesquiterpenoids compared to the non-transgenic root lines [95].

Besides single-gene manipulation, the integration of heterologous genes and entire biosynthetic pathways into hairy root cultures through synthetic biology has been employed to redirect native metabolic flux. Likewise, the production of benzylisoquinoline alkaloids [96,97,98] and abietane diterpenoids [99] was enhanced via metabolic engineering in hairy root cultures. A metabolic engineering approach involving the upregulation of the *CHSa* gene from Petunia has been used to enhance flavonolignan biosynthesis in the hairy root cultures of *Silybum marianum*. As a result, these transgenic roots exhibited a seven-fold and ten-fold increase in silymarin and silybin content, respectively, compared to the non-transgenic hairy roots [100]. Another notable example involves the fusion of upstream and downstream biosynthetic enzymes in *Taxus chinensis* hairy roots, which facilitated efficient substrate channeling and resulted in a 5-fold increase in paclitaxel biosynthesis [101]. This strategy minimizes intermediate diffusion and reduces metabolic by-products, thereby improving overall pathway efficiency. Similarly, in *Catharanthus roseus*, the fusion of tryptophan decarboxylase (TDC) and strictosidine synthase (STR) enhanced the biosynthesis of monoterpene indole alkaloids by 3.5-fold through improved substrate channeling [102,103]. Furthermore, the metabolic engineering of tropane alkaloid pathway in hairy root cultures of the native Tibetan medicinal plant *Scopolia lurida* was achieved through the heterologous expression of the *Hyoscyamus niger hyoscyamine 6-hydroxylase gene* (*HnH6H*) and the homologous expression of the *S. lurida H6H* gene (*SlH6H*). This modification significantly enhanced the accumulation of scopolamine and anisodamine while reducing hyoscyamine levels. The *HnH6H*-overexpressing hairy root cultures produced more scopolamine than those overexpressing *SlH6H*, demonstrating the superior efficiency of HnH6H in enhancing scopolamine biosynthesis.

### 4.2. Scaling Metabolite Production via Bioreactor Systems

Despite the demonstrated success of hairy root cultures at the laboratory scale, their transition to industrial applications necessitates the optimization of growth conditions and efficient metabolite extraction methods [73]. Prior to large-scale bioreactor application, laboratory-scale systems such as Erlenmeyer flasks on orbital shakers and benchtop bubble column reactors are commonly employed to establish foundational growth parameters. Orbital shaker parameters, including shaking speed (typically 100–120 rpm), temperature (22–25 °C), and light/dark conditions, significantly influence biomass accumulation and metabolite output. For example, in hairy roots of *Hyoscyamus muticus*, optimal growth was achieved at 110 rpm and 24 °C under dark conditions, resulting in enhanced tropane alkaloid synthesis [58]. This study utilized orbital shaker conditions of 100 rpm at 25 °C in darkness to optimize tropane alkaloid production in *H. muticus* hairy root cultures. Moreover, disposable culture vessels with controlled aeration and autoclavable mini-bioreactors (1–2 L capacity) have enabled efficient root proliferation and preliminary scaling while maintaining sterility and reproducibility [104,105]. These systems act as a bridge between static flask cultures and advanced bioreactor designs.

Bioreactor systems have proven instrumental in addressing scalability challenges, offering enhanced control over environmental parameters and improved productivity [106]. Airlift bioreactors, for example, have demonstrated significant improvements in oxygen transfer and nutrient distribution, which are critical for supporting high-density hairy root growth [107,108]. In *Datura stramonium*, the implementation of a modified stirred tank reactor maintained 95% root viability while achieving a 20-fold increase in hyoscyamine production compared to traditional flask cultures [109]. Similarly, three-dimensional (3D)-printed scaffold reactors have emerged as innovative tools for facilitating spatially controlled root growth and enhancing metabolite production [110]. In *C. roseus*, the use of 3D-printed scaffolds led to a 30% increase in vincristine production by optimizing root architecture and nutrient access [110]. These systems enable precise manipulation of the root environment, leading to improved yields in valuable specialized metabolites.

Advancements in continuous-fed bioreactors have further enhanced metabolite yields by enabling sustained nutrient replenishment and in situ extraction of bioactive compounds [111]. For example, a perfusion-based bioreactor system for *G. uralensis* hairy roots resulted in a 10-fold increase in glycyrrhizin accumulation while preventing product inhibition [112]. This system allows continuous removal of the target metabolite, reducing feedback inhibition and maintaining high productivity over extended periods. Additionally, co-culture strategies integrating hairy roots with engineered microbial strains have been explored for enhanced metabolite production. For example, the co-cultivation of plants with metabolically engineered yeast facilitated a 25-fold increase in resveratrol production through cross-kingdom metabolic coupling [113,114]. This approach combines the biosynthetic capabilities of plant and microbial systems, offering new avenues for producing complex metabolites that are difficult to synthesize using traditional methods. Further innovations in bioreactor technology include the development of wave-mixed bioreactors, which provide gentle mixing and aeration, reducing shear stress on hairy roots and promoting uniform growth (Table 2). For instance, wave-mixed bioreactors have been used to enhance ginsenoside production in *P. ginseng*, achieving a 15-fold increase in yield compared to shake flask cultures [115]. These advancements in bioreactor systems highlight their crucial role in scaling up hairy root cultures for industrial production of high-value metabolites, offering a sustainable alternative to traditional plant-based extraction methods.

### 4.3. Elicitation Strategies in Hairy Root Culture to Enhance Metabolite Biosynthesis

Elicitation strategies in hairy root culture systems have emerged as powerful tools to enhance the biosynthesis of specialized metabolites, addressing the challenges of low yields and complex extraction processes in medicinal plants [70]. By applying biotic and abiotic elicitors, researchers can stimulate plant defense responses and upregulate key biosynthetic pathways, signifcantly increasing the production of high-value secondary metabolites. The following subsections provide a detailed overview of these elicitation approaches, highlighting their mechanisms and impacts on metabolite production in hairy root culture systems.

#### 4.3.1. Organic and Biotic Elicitors

The application of organic and biotic elicitors in hairy root cultures has significantly impacted secondary metabolite production. Among organic elicitors, methyl jasmonate has proven highly effective in enhancing secondary metabolite yields across diverse plant species. For instance, treatment of *Panax ginseng* hairy roots with methyl jasmonate and cellulose nanocrystals led to a 12-fold increase in ginsenoside accumulation [116,120]. This enhancement was attributed to the upregulation of cytochrome P450 and glycosyltransferase genes, which are crucial for ginsenoside biosynthesis. Similarly, in *Panax quinquefolium*, methyl jasmonate elicitation substantially enhanced ginsenoside content from 8.32 mg/g to 43.66 mg/g [121]. *Hyoscyamus niger* hairy root cultures treated with methyl jasmonate showed notable enhancement in tropane alkaloid biosynthesis, specifically increasing scopolamine yield significantly [122]. Salicylic acid, another potent organic elicitor, notably elevated the production of alkaloids such as vinblastine and vincristine by approximately two-fold in *C. roseus* hairy roots by upregulating biosynthetic pathways and associated enzymes [123].

The use of biotic elicitors, derived from microorganisms or plant pathogens, are frequently applied to mimic pathogen attack and trigger plant-specialized metabolic defense responses. One of the prominent biotic elicitors is yeast extract, which has been demonstrated to notably enhance the production of secondary metabolites such as rosmarinic acid in *S. miltiorrhiza* hairy root cultures. For instance, a previous study reported that yeast extract augmented the expression of key enzymes within the biosynthetic pathways, thereby stimulating metabolite accumulation [124]. The combined treatment of yeast extract with other elicitors can further amplify yield, as evidenced by studies showing that such treatments enhance metabolite yields beyond what each elicitor achieves alone [125]. Chitosan, another notable biotic elicitor, has shown remarkable potential in enhancing the production of artemisinin in *A. annua* hairy roots. Chitosan elicits a stress response that activates pathways associated with secondary metabolite biosynthesis [124]. Research indicates that the use of chitosan not only increased the overall yield but also improved the growth of the hairy root cultures, thus providing a dual benefit for biotechnological applications [126]. Furthermore, fungi-derived biotic elicitors such as endophytes have emerged as promising candidates for enhancing secondary metabolite production. For example, the endophyte *Chaetomium globosum* was shown to significantly stimulate tanshinone accumulation in *S. miltiorrhiza* hairy root cultures. Researchers demonstrated that the biotic interactions between the endophyte and plant led to increased biosynthesis of desired metabolites, further emphasizing the intricate relationships within the plant–microbe ecosystem [127].

#### 4.3.2. Abiotic Elicitors

Abiotic elicitors are non-living physical or chemical factors that induce physiological and metabolic responses in plants, often mimicking environmental stress signals. In hairy root culture systems, these elicitors are strategically applied to stimulate the biosynthetic machinery responsible for the production of specialized metabolites. Common abiotic elicitors include signaling molecules such as methyl jasmonate, salicylic acid, silver nitrate (AgNO_3_), and hydrogen peroxide (H_2_O_2_), as well as heavy metals, UV radiation, and osmotic agents [100,101].
(a)Physical Elicitors

Physical elicitors such as light modulation, ultrasound, and UV radiation have been widely utilized to trigger stress responses in hairy root systems. For example, blue light exposure in *Hypericum perforatum* hairy roots triggered a 10-fold increase in hypericin accumulation [118]. This enhancement was mediated via photoreceptor-regulated activation of polyketide synthases, essential enzymes in the hypericin biosynthesis pathway [118]. Likewise, UV-B radiation has been reported to elevate flavonoid biosynthesis in *Glycyrrhiza glabra* hairy roots by upregulating genes in the phenylpropanoid pathway [76]. Another notable physical elicitor is ultrasound treatment. In *Artemisia annua*, ultrasonic waves enhanced artemisinin biosynthesis 2.5-fold through mechanical stress-induced activation of defense-related pathways [128].
(b)Chemical Elicitors

Chemical abiotic elicitors include nanoparticles, metal ions, and osmotic agents. Recent advances in nanotechnology have enabled precise modulation of plant metabolic pathways. For instance, zinc oxide nanoparticles (ZnO-NPs) have been shown to significantly enhance vinblastine production in *C. roseus* hairy roots by inducing oxidative stress and activating related gene networks [119]. Other chemical elicitors such as copper sulfate (CuSO_4_) and sodium chloride (NaCl) have also been tested for their role in modulating biosynthetic flux in hairy root systems, although further optimization is often needed for species-specific responses.

These examples underscore the diversity and potential of both biotic and abiotic elicitors in enhancing the production of valuable specialized metabolites in hairy root cultures. To fully leverage these strategies, careful timing of elicitor application, concentration optimization, and synergistic use of elicitor combinations are critical factors influencing the magnitude of the response

#### 4.3.3. Inducible Metabolites Secreted into the Culture Medium of Hairy Roots

Hairy root cultures are a versatile platform for producing inducible metabolites secreted into the culture medium, with elicitation strategies significantly enhancing yields. These extracellularly secreted compounds are generally absent under basal conditions and are only detected after the application of specific biotic or abiotic elicitors, offering a scalable and purification-friendly alternative to intracellular extraction. This phenomenon is of great scientific and industrial interest because it simplifies downstream purification processes and facilitates scalable production of bioactive compounds for pharmaceutical, cosmetic, and food industries. Below is a comprehensive summary of key findings from published studies on inducible metabolites secreted into the culture medium of hairy root cultures, organized by metabolite class and plant species (Table 3).

##### Alkaloids

Alkaloids are nitrogen-containing secondary metabolites with profound pharmaceutical value. Hairy root cultures have been effectively utilized to enhance the production and secretion of various alkaloids upon elicitation. *C. roseus* is a prominent model for producing terpenoid indole alkaloids (TIAs) such as ajmalicine, serpentine, and catharanthine. Elicitation with methyl jasmonate or fungal extracts like *A. niger* significantly enhances extracellular accumulation [129]. Previous studies demonstrated a 5-fold increase in alkaloid secretion when *A. niger* elicitation was combined with in situ adsorption using Amberlite XAD-7 [129,130]. Similarly, *N. tabacum* hairy root cultures respond to MeJA or salicylic acid (SA) treatment by enhancing nicotine secretion [131], reporting a 2-fold increase attributed to the upregulation of putrescine N-methyltransferase (PMT) genes. In *Brugmansia candida*, hairy root cultures produce tropane alkaloids like hyoscyamine and scopolamine [132]. Elicitation with biotic agents such as yeast extract or abiotic stressors like silver nitrate enhances secretion, observing a 3-fold increase in extracellular scopolamine levels following yeast extract treatment [132].

##### Phenylpropanoids

Phenylpropanoids encompass a wide array of compounds including phenolics and flavonoids, known for their antioxidant and medicinal properties. Hairy root cultures of *Mentha pulegium* (pennyroyal) have shown substantial production of rosmarinic acid and other phenolic compounds, particularly upon elicitation with MeJA or *A. rhizogenes*, with a 2.5-fold increase in extracellular phenolic content [133]. *Tetrastigma hemsleyanum*, a medicinal plant rich in flavonoids, shows increased secretion of luteolin upon treatment with MeJA or chitosan. A 3-fold increase in flavonoid accumulation in the culture medium was observed, with luteolin reaching 1.2 mg/L [134]. Likewise, in *Erigeron breviscapus*, hairy root cultures produce the flavone scutellarin, which exhibits cardiovascular benefits. Elicitation with jasmonic acid or UV radiation significantly improves secretion, with a 4-fold increase in extracellular scutellarin, thus streamlining downstream processing [135].

##### Terpenoids

Terpenoids such as sesquiterpenes and diterpenes hold great pharmaceutical and agricultural importance. In *A. annua*, hairy root cultures are capable of producing artemisinin, a vital antimalarial drug. Elicitation with silver nanoparticles or MeJA enhances secretion, with a 3-fold increase, reaching 0.8 mg/L in the culture medium [136]. Solanum tuberosum (potato) hairy roots secrete antimicrobial sesquiterpenes such as rishitin upon elicitation with fungal homogenates, particularly from Phytophthora infestans, observing a 2.5-fold increase [137]. In Tanacetum parthenium, parthenolide—a sesquiterpene lactone with anti-inflammatory properties—is secreted into the medium upon combined MeJA and UV radiation treatment [138]. This resulted in a 3.5-fold enhancement in extracellular parthenolide content under these elicitation conditions [138].

##### Anthraquinones

Anthraquinones are aromatic compounds widely used in pharmaceuticals and natural dyes. Hairy root cultures of *Polygonum multiflorum* synthesize and secrete emodin and related anthraquinones when treated with MeJA or SA [139]. A 2-fold increase was observed in extracellular emodin, achieving concentrations of 0.5 mg/L [139]. Similarly, *Rubia* species have demonstrated the ability to produce and secrete anthraquinone derivatives upon fungal or heavy metal elicitation [140]. They observed that fungal extracts from *Aspergillus* significantly enhanced anthraquinone levels 2.8-fold in the culture medium [140].

##### Recombinant Proteins

Hairy root cultures are not limited to secondary metabolites; they are also valuable platforms for recombinant protein production. In *N. tabacum* and *N. benthamiana*, proteins such as thaumatin, human erythropoietin (rhEPO), and acetylcholinesterase (AChE) have been successfully secreted into the medium. Thaumatin secretion was enhanced 3-fold by polyvinylpyrrolidone (PVP) supplementation [141]. Similarly, a prior work optimized conditions that led to a 3-fold increase in AChE secretion, which reached 3.3% of total soluble protein in the medium [142]. Furthermore, the secretion of the murine IgG1 monoclonal antibody at 18 mg/L in *N. tabacum* hairy roots was increased by up to 43% with PVP and gelatin supplementation, underlining the potential of this system for biopharmaceutical production [143].

**Table 3 plants-14-01928-t003:** Representative examples of inducible metabolites secreted into the culture medium of hairy roots upon elicitation.

Plant Species	Elicitor Used	Metabolite(s)	Compound Class	Fold Increase/Yield	Reference
*Hyoscyamus muticus*	AgNPs	Scopolamine	Tropane alkaloid	~4-fold increase (media)	[102]
*Rubia cordifolia*	MeJA	Anthraquinones	Quinones	Significant	[140]
*Plumbago rosea*	Ag^+^, MeJA	Plumbagin	Naphthoquinone	~3-fold (extracellular)	[144]
*Coptis japonica*	Yeast extract	Berberine	Isoquinoline alkaloid	Not quantified	[145]
*Glycyrrhiza uralensis*	UV-C, Chitosan	Glycyrrhizin	Triterpenoid saponin	Up to 10-fold (medium)	[146]
*Salvia miltiorrhiza*	MeJA	Tanshinones	Diterpenoids	Increased secretion	[147]
*Panax ginseng*	MeJA, CNCs	Ginsenosides	Triterpenoid saponins	12-fold increase (total)	[148]
*Beta vulgaris*	AgNPs	Betalains	Alkaloids	15-fold (culture media)	[117]
*Catharanthus roseus*	ZnO-NPs	Vinblastine	Indole alkaloid	4.3-fold (media)	[149]
*Silybum marianum*	CuSO_4_	Silymarin, Silybin	Flavonolignans	7–10× (media)	[150]
*Artemisia annua*	NO_3_^−^/NH4^+^	Artemisinin	Sesquiterpenoid	2.5× (extracellular)	[151]

## 5. Challenges and Limitations of Hairy Root Culture for Specialized Metabolite Production

Hairy root cultures have emerged as promising biotechnological tools for the production of high-value specialized metabolites. However, their commercial and industrial application remain constrained by several critical challenges. These include limitations in bioreactor scalability, specificity of metabolite biosynthesis, long-term genetic and metabolic stability, and reproducibility of culture performance under varied conditions. A detailed analysis of these limitations is crucial to moving from laboratory-scale research to viable industrial platforms.

### 5.1. Bioreactor Design and Scale-Up Constraints

The primary limitation of hairy root culture is their difficulty in scaling up production while maintaining metabolic efficiency. Bioreactor systems, although essential for large-scale cultivation, present significant technical challenges [152,153]. Unlike microbial or suspension cell cultures, hairy roots have low shear resistance, making them highly sensitive to mechanical forces exerted by vigorous mixing or aeration, which are necessary for efficient nutrient and oxygen distribution [153]. Suboptimal oxygen transfer and nutrient gradients often result in heterogeneity within the culture, leading to inconsistent growth and metabolite yields [154]. To overcome these limitations, a range of bioreactor configurations has been investigated. For instance, airlift bioreactors enhanced oxygen distribution while minimizing shear stress in *Hyoscyamus muticus* hairy roots [152,153], and wave-induced bioreactors improved biomass and atropine content in *Datura* roots [155] (Figure 3). Nonetheless, these systems still require high capital investment, rigorous contamination control, and real-time monitoring of physical and biochemical parameters, which remain a major bottleneck in industrial applications [156]. Additionally, metabolite accumulation in the culture medium can exert feedback inhibition, necessitating strategies such as in situ product removal to prevent toxicity and metabolic repression [112].

### 5.2. Metabolite Specificity and Biosynthetic Limitations

Although hairy root cultures exhibit a remarkable ability to synthesize specialized metabolites, their utility is inherently restricted by tissue-specific biosynthesis patterns. Some high-value compounds are predominantly synthesized in aerial parts of plants, such as leaves, stems, or flowers, rather than in roots [157]. For instance, certain flavonoids and terpenoids, despite being successfully synthesized in hairy roots, require additional enzymatic modifications that are naturally carried out in other plant tissues, leading to incomplete biosynthesis or reduced bioactivity [158]. For example, taxol precursors are limited in root tissues and require transformation of cell lines with additional P450 monooxygenases to achieve complete biosynthesis [159]. In the same way, monoterpenes such as linalool and geraniol in *Lavandula angustifolia* are primarily localized in glandular trichomes rather than roots, limiting their expression in hairy root cultures unless metabolic pathway engineering is applied [160,161,162].

### 5.3. Elicitation Variability and Culture Reproducibility

Furthermore, while elicitors such as jasmonic acid, salicylic acid, and nanoparticles have been shown to enhance metabolite production, the long-term stability and reproducibility of these induction strategies remain a challenge. For instance, MeJA elicitation increased ginsenoside production in *P. ginseng*, but the yield declined significantly after five subculturing cycles [148]. In *W. somnifera*, elicitor-triggered withanolide synthesis was shown to be affected by light and passage number, indicating epigenetic and environmental interactions [163].

### 5.4. Genetic Stability and Long-Term Preservation Issues

Moreover, preservation and long-term maintenance of hairy root cultures present another major bottleneck, particularly for industrial applications that require continuous production [164]. Hairy roots cannot be stored as seed stocks; they require cryopreservation or routine subculture, both of which are labor-intensive and risk introducing somaclonal variation or loss of metabolite productivity [165]. Cryopreservation techniques, including slow freezing, vitrification, and encapsulation–dehydration, have been explored to store hairy roots for extended periods, yet maintaining their genetic stability and post-thaw regeneration capacity remains challenging [166]. The accumulation of oxidative stress during storage and recovery further complicates long-term viability, necessitating the use of antioxidant treatments or stress-protective agents to improve survival rates. Additionally, repeated subculturing of hairy roots in vitro often leads to somaclonal variation, epigenetic drift, or metabolic shifts that alter specialized metabolite profiles, making it difficult to maintain consistent production over extended periods [155].

Addressing these challenges requires an integrative approach that combines advances in bioreactor design, metabolic engineering, preservation techniques, and stress adaptation strategies. The future of hairy root culture lies in leveraging synthetic biology, machine learning-based optimization, and innovative cultivation strategies to overcome current limitations and unlock their full potential as a sustainable platform for specialized metabolite production and beyond.

## 6. Emerging Technologies and Future Directions

Recent advancements in artificial intelligence (AI)-guided metabolic engineering and synthetic biology have further expanded the capabilities of hairy root cultures [167,168]. Machine learning algorithms, such as the DeepMetabolite platform, are being utilized to predict optimal gene combinations for metabolite overproduction, reducing the need for traditional trial-and-error approaches [169]. CRISPR-Combo systems, which integrate gene editing with transcriptional activation, have been successfully applied in *C. roseus* to enhance alkaloid biosynthesis by 40% [170]. Similarly, synthetic microbial consortia are also gaining attraction as a novel strategy for improving metabolite yields. By co-culturing hairy roots with engineered microbial strains, researchers have achieved enhanced metabolite synthesis through interspecies metabolic exchange [171]. Additionally, advances in single-cell sequencing and spatial transcriptomics are enabling a deeper understanding of cell-type-specific metabolic variations within hairy root cultures, paving the way for more precise metabolic engineering strategies.

Besides pharmaceuticals and nutraceuticals, hairy root cultures are being explored for applications in sustainable agriculture and environmental biotechnology [21]. Engineered hairy roots of *Brassica juncea* and *N. tabacum* have shown enhanced heavy metal uptake, demonstrating their potential for phytoremediation. Furthermore, metabolic engineering of *Azadirachta indica* hairy roots has led to increased azadirachtin production, a potent botanical insecticide with applications in integrated pest management [172]. In summary, hairy root cultures offer a valuable biotechnological platform for the controlled and scalable production of plant-based compounds. By integrating metabolic engineering, elicitation strategies, bioreactor innovations, and AI-assisted process optimization, researchers are progressively enhancing their efficacy. Advancements in synthetic biology and omics technologies, such as transcriptomics and metabolomics, are contributing to the targeted enhancement of biosynthetic pathways, further positioning hairy root cultures as a crucial tool in pharmaceutical, agricultural, and environmental biotechnology. Although challenges remain, their potential for addressing specific production bottlenecks in natural compound supply chains is becoming increasingly evident.

## 7. Conclusions

Hairy root cultures have emerged as a robust and versatile platform for the sustainable production of plant-derived bioactive compounds, specialized metabolites, and recombinant proteins. Their ability to maintain genetic and biochemical stability, coupled with high metabolite productivity and hormone-independent growth, makes them an attractive system for both fundamental research and industrial biotechnology.

Despite these advantages, the transition from laboratory-scale systems to industrial applications remains hindered by several challenges, including scale-up limitations, metabolite specificity to aerial plant parts, variability in elicitor responsiveness, and long-term maintenance of biosynthetic consistency. Recent developments in bioreactor technologies—such as wave-mixed, airlift, and 3D-printed scaffold systems—have significantly improved root biomass and metabolite yields while mitigating physical stress and nutrient heterogeneity. However, challenges related to oxygen transfer, metabolite feedback inhibition, and cost-effective culture preservation remain critical areas for improvement.

At the same time, the integration of elicitation strategies, metabolic pathway engineering, and synthetic biology approaches is opening new possibilities to overcoming tissue-specific limitations and enhance specialized metabolite output. For instance, the co-expression of key pathway genes, CRISPR/Cas-mediated genome editing, and the use of abiotic elicitors like nanoparticles and light wavelength modulation have demonstrated promising results. Furthermore, the adoption of omics-based tools (e.g., transcriptomics, metabolomics, and proteomics) is enabling a more precise understanding of the molecular basis of metabolite biosynthesis in hairy root cultures.

Looking forward, the incorporation of artificial intelligence (AI) and machine learning algorithms for optimizing culture conditions, elicitor regimes, and predictive metabolite modeling is poised to accelerate the development of HRCs as scalable, cost-effective, and reliable production systems. Moreover, innovative applications beyond pharmaceuticals—such as phytoremediation, pest management, and nutraceutical production—underscore the multi-dimensional utility of HRCs in addressing global challenges.

In conclusion, while several biological and technological hurdles remain, hairy root cultures represent a maturing biotechnological tool with immense potential for future applications across pharmaceutical, agricultural, and environmental sectors. Continued interdisciplinary efforts combining plant biotechnology, bioengineering, data science, and systems biology will be essential to fully harness the capabilities of this system and translate its promise into tangible socio-economic and ecological benefits.

## Figures and Tables

**Figure 1 plants-14-01928-f001:**
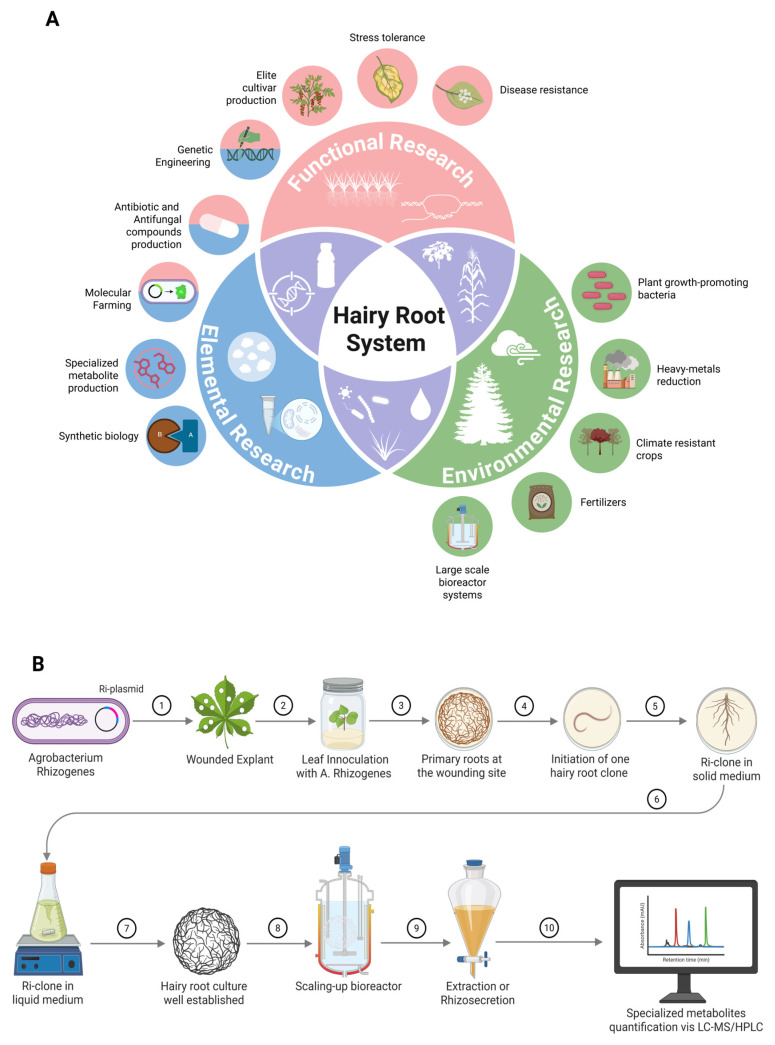
Applications and the establishment of hairy root culture system for large-scale production of plant-specialized metabolites. (**A**) The illustration of key hairy root applications such as biosynthesis of recombinant proteins and specialized metabolites, transgenic plant research for stress resistance, and environmental remediation. (**B**) Hairy root culture is initiated by infecting a wounded leaf with *Agrobacterium rhizogenes* carrying either wild-type or engineered T-DNA (step 1-2). After 48–72 h of co-culture, adventitious roots emerge near the infection site (step 3-4). These roots are excised and transferred to a solid medium to induce clonal hairy root formation (step 5). Root tips are then subcultured in an agitated liquid medium for further proliferation (step 6). A well-established hairy root clone exhibits vigorous, plagiotropic growth with extensive branching under hormone-free conditions (step 7). The culture is subsequently scaled up in a bioreactor to facilitate terpenoid production (step 8), and the resulting compounds are identified and quantified using LC-MS or HPLC analysis (step 9-10).

**Figure 2 plants-14-01928-f002:**
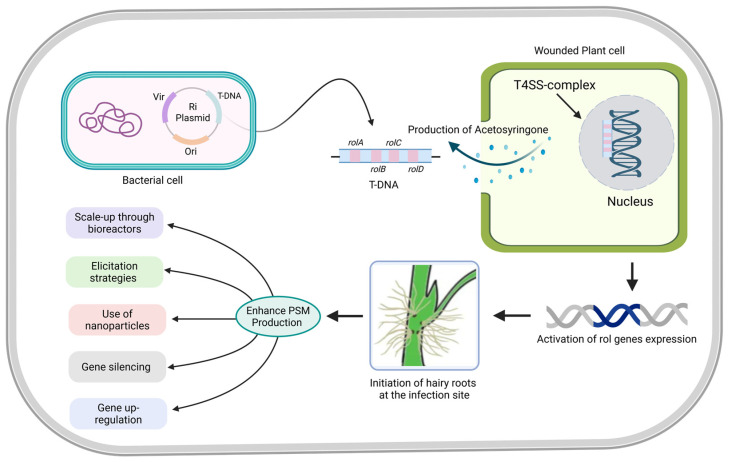
Schematic representation of the molecular mechanism of *A. rhizogenic*-mediated hairy root induction in plants. The process begins with the introduction of a recombinant plasmid carrying an exogenous gene into *A. rhizogens*. Upon wounding of plant explants, phenolic compounds and other signaling molecules released from the damaged tissue activate the bacterial *vir* genes, initiating the transfer of T-DNA from the Ri plasmid into the host plant cells. The T-DNA integrates into the plant genome, leading to the expression of *rol* and auxin-related genes, which disrupt endogenous hormonal balance and promote the formation of genetically transformed hairy roots. This transformation is further facilitated by bacterial signaling molecules and plant growth regulators, resulting in enhanced root proliferation. The establishment of hairy roots in plants serve as valuable platforms for metabolic engineering, pathway elucidation, and functional genomics. Scalable in bioreactors, these cultures enable sustainable production of high-value compounds and can be used to generate rare or novel metabolites under controlled conditions.

**Figure 3 plants-14-01928-f003:**
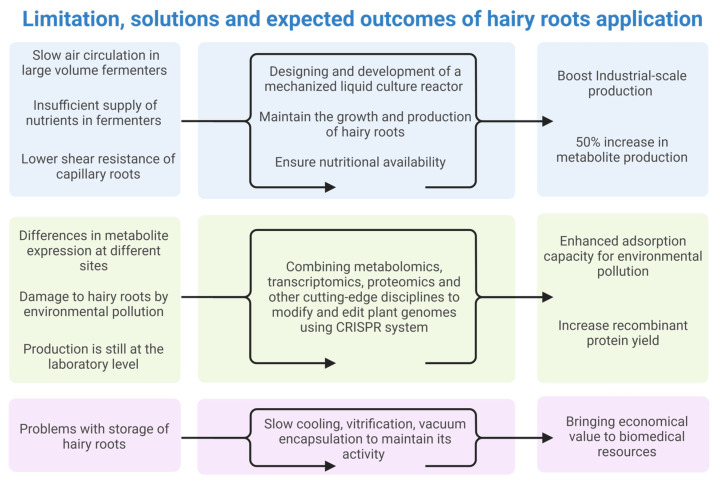
Overcoming the limitations of the hairy root applications for industrial use. This figure illustrates the key limitations, proposed solutions, and predicted outcomes of applying hairy roots in large-scale industrial and environmental contexts. The left panel outlines the primary challenges faced in cultivating and using hairy roots, including issues such as slow air circulation in large fermenters, insufficient nutrient supply, low shear resistance, variable metabolite expression, vulnerability to environmental pollutants, and challenges with scalability and storage. The middle panel presents proposed solutions aimed at overcoming these limitations. These include the development of mechanized liquid culture reactors to enhance growth and nutrient supply, the integration of advanced bioinformatics and gene editing techniques like CRISPR for optimized metabolite production, and the improvement of preservation techniques to maintain the biological activity of hairy roots. The right panel highlights the anticipated outcomes of these solutions, including industrial-scale production, increased metabolite and recombinant protein yields, enhanced environmental pollution adsorption, and contributions to addressing medicinal resource shortages. Additionally, these advancements are expected to generate significant economic benefits to the biomedical sector, ultimately driving innovation in healthcare and environmental sustainability.

**Table 1 plants-14-01928-t001:** The enhancement of key bioactive compounds produced by hairy root cultures using different strategies in a variety of industrially important plant species.

Plant Species	Metabolite	Business Value	Strategy	Yield	Reference
Tetrastigma hemsleyanum	Catechin	Anti-tumor, antioxidant	Hormones-induced co-culture time expansion	692.63 ± 127.24 mg/g DW	[43]
	Epicatechin	Antioxidant, lipid glucose-lowering		163.34 ± 31.86 mg/g DW	[43]
*Stephania tetrandra*	Tetrandrine	Anti-rheumatic, anti-inflammatory	Use of WPM medium	7.28 mg/L DW	[79]
	Total phenolics	Antioxidant, anti-inflammatory		7.28 mg/L DW	[79]
*Trigonella* *foenum graecum*	Total phenolics	Antioxidant, anti-inflammatory	Processing with SA	15.082 ± 1.211 μg/DW	[80]
	Flavanol	Antiallergic		18.587 ± 2.564 μg/DW	
	Flavonoids	Antioxidant, anti-inflammatory		15.082 ± 1.211 μg/DW	
	Anthocyanin	Antioxidant, anti-inflammatory		2.727 ± 0.076 μg/DW	
*Calendula officinalis*	Saponins	Antibacterial, antiviral	Triton X-100 addition	1.2 mg/g DW	[81]
Aucher ex Benth.	Aalvianolic acid	Antibacterial	Processing with Ag^+^	31.49 ± 0.65 mg/L DW	[82]
*Withania somnifera*	Lactones	Anti-inflammatory, anti-tumor	Elicitor treatment with MeJA and β-CD	17.45 mg/g DW	[83]
*Cannabis sativa*	Friedelin	Antidiabetic, hypolipidemic	Processing with SA	5.018 ± 0.35 mg/g DW	[84]
*Cannabis sativa*	*Epifriedelanol*	Antidiabetic, hypolipidemic	Processing with SA	5.018 ± 0.35 mg/g DW	[84]
*pigeon pea*	Flavonoids	Antiallergic	UV-B radiation	414.95 ± 50.68 μg/g DW	[85]
	Cajaninstilbenate	Antioxidant	UV-B radiation	666.01 ± 702.14 μg/g DW	[85]

**Table 2 plants-14-01928-t002:** Major advancements in metabolic engineering of hairy root cultures for the enhanced biosynthesis of plant-derived natural products.

Application	Plant Species	Biological Impact	Reference
Metabolic engineering for pathway optimization	*Artemisia annua*	CRISPR-Cas9 knockout of squalene synthase (SQS) redirected precursor flux, leading to a 3.2-fold increase in artemisinin production.	[90]
	*Salvia miltiorrhiza*	Overexpression of SmMYB1 activated diterpenoid biosynthetic genes, resulting in an 8.5-fold increase in tanshinone production.	[91]
	*Catharanthus roseus*	Introduction of strictosidine synthase enabled de novo production of monoterpene indole alkaloids.	[102]
	*Withania somnifera*	Co-overexpression of squalene epoxidase and cytochrome P450 genes significantly enhanced withanolide accumulation.	[92]
	*Taxus chinensis*	Fusion of biosynthetic enzymes facilitated efficient substrate channeling, increasing paclitaxel biosynthesis 5-fold.	[101]
Elicitation-induced biosynthesis	*Panax ginseng*	Treatment with methyl jasmonate (MeJA) and cellulose nanocrystals resulted in a 12-fold increase in ginsenoside production.	[116]
	*Beta vulgaris*	Silver nanoparticles (AgNP) stimulated betalain production 15-fold via ROS-mediated pathway activation.	[117]
	*Hypericum perforatum*	Blue light exposure triggered a 10-fold increase in hypericin accumulation by activating polyketide synthases.	[118]
	*Glycyrrhiza glabra*	UV-B irradiation enhanced flavonoid biosynthesis by inducing phenylpropanoid pathway genes.	[76]
	*Catharanthus roseus*	Zinc oxide nanoparticles (ZnO-NPs) enhanced vinblastine production by modulating oxidative stress responses.	[119]
Scaling Production via Bioreactor Systems	*Datura stramonium*	Stirred tank reactor maintained 95% root viability and increased hyoscyamine production.	[109]
	*Catharanthus roseus*	3D-printed scaffold reactors improved vincristine production by 30%.	[110].
	*Glycyrrhiza uralensis*	Perfusion-based bioreactor resulted in 10-fold increase in glycyrrhizin accumulation while preventing product inhibition.	[112]
	*Polygonum cuspidatum*	Co-culture with *Saccharomyces cerevisiae* enabled 25-fold increase in resveratrol production.	[113,114].

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
