# Peer review of "Reprogramming Hairy Root Cultures: A Synthetic Biology Framework for Precision Metabolite Biosynthesis"

_plants, 2025, doi:10.3390/plants14131928_

Round 1

Reviewer 1 Report

Comments and Suggestions for Authors

The authors describe hairy roots as a sustainable bioproduction platform for valuable plant-specialized metabolites. Hairy root cultures represent a valuable source for producing bioactive metabolites that are present in only minute quantities in intact plants. While the review is well-written, I have several comments and recommendations that should be addressed before publication consideration.

  1. Figures 1, 2, and 3 appear overly stretched. Please correct their dimensions for better readability.
  2. Please italicize bacterial names and gene names throughout the manuscript. Currently, they are not italicized.
  3. The manuscript should discuss the genetic differences and transformation efficiency among commonly used bacterial strains for hairy root induction such as ATCC 15834, K599, etc.
  4. Lines 362 and 374: Please delete the citations Rahnama et al., 2013 and Lan et al., 2018, and provide reference numbers for these citations instead.
  5. The authors should include a comprehensive table summarizing different hairy root cultures with enhanced metabolite production under various elicitor treatments. Additionally, authors should briefly describe the role of commonly used elicitors for enhancing metabolite production from hairy root cultures.
  6. Several studies report significantly enhanced metabolite production in hairy root cultures. However, the isolation of such inducible metabolites in large quantities from hairy root culture media is not commonly reported. A few publications describe the isolation of plant-specialized metabolites in amounts reaching hundreds of milligrams from elicited hairy root cultures. A representative example is the work of Sharma et al. 2022, "Induction of the prenylated stilbenoids arachidin-1 and arachidin-3 and their semi-preparative separation and purification from hairy root cultures of peanut (Arachis hypogaea)", where researchers isolated prenylated stilbenes from hairy root cultures of peanut in large amounts, demonstrating the potential of hairy roots as a sustainable bioproduction platform.
  7. Please consider using plant-specialized metabolites instead of secondary metabolites for accuracy.
  8. The authors should elaborate on why hairy roots are essential for metabolite production, particularly addressing challenges like low yield and seasonal variation in field-collected plants.
  9. Hairy roots typically do not produce metabolites in liquid culture without elicitation. The authors should include a table summarizing enhanced metabolite production under different biotic and abiotic elicitors. Since inducible metabolites in liquid culture media are more scientifically valuable than constitutive metabolites present in the tissue itself, I recommend focusing particularly on inducible metabolites produced by hairy root cultures. Representative examples include prenylated stilbenes from peanut (Arachis hypogaea) hairy root cultures, stilbenes and benzofurans from mulberry (Morus) hairy root cultures, and geranylated flavonoid from Dalea hairy root cultures.
  10. Please briefly describe the uses of hairy root cultures in addition to metabolite production.
  11. Please elaborate on trichome root culture or, if more appropriate, replace this term with hairy root cultures as used in Figure 3.

Author Response

Dear Editor and Reviewers,

We sincerely thank you for the thorough evaluation of our manuscript, "Advancing Hairy Root Culture Systems for Sustainable and Scalable Production of Plant-Specialized Metabolites." We are grateful for your insightful comments and constructive suggestions, which have helped us significantly improve the clarity, scientific depth, and overall quality of the manuscript. We have carefully addressed each point raised by the reviewers and made the corresponding revisions in the manuscript. Below, we provide a detailed point-by-point response, highlighting the changes made and offering clarifications where necessary. All modifications have been marked in the revised version of the manuscript for your convenience.

Reviewer 1

We sincerely thank Reviewer 1 for the thoughtful and constructive feedback. Below, we address each comment in detail and explain the corresponding changes made to the manuscript.

1. Figures 1, 2, and 3 appear overly stretched. Please correct their dimensions for better readability.

Response:
We appreciate the reviewer’s observation. Figures 1, 2, and 3 have been reformatted to maintain proper aspect ratios and improve visual clarity and readability in the revised manuscript.

2. Please italicize bacterial names and gene names throughout the manuscript. Currently, they are not italicized.

Response:
Thank you for pointing this out. All bacterial names (e.g., Rhizobium rhizogenes, Agrobacterium tumefaciens) and gene names have now been consistently italicized throughout the manuscript, in accordance with standard scientific conventions.

3. The manuscript should discuss the genetic differences and transformation efficiency among commonly used bacterial strains for hairy root induction such as ATCC 15834, K599, etc.

Response:
As suggested, we have added a dedicated paragraph discussing the genetic characteristics and transformation efficiency of frequently used R. rhizogenes strains (e.g., ATCC 15834, K599, LBA9402). This information has been included in the revised section on "Hairy Root Induction and Transformation Parameters" (Section 3.1).

4. Lines 362 and 374: Please delete the citations Rahnama et al., 2013 and Lan et al., 2018, and provide reference numbers for these citations instead.

Response:
We have replaced the in-text names with their corresponding reference numbers in accordance with the journal’s citation style.

5. The authors should include a comprehensive table summarizing different hairy root cultures with enhanced metabolite production under various elicitor treatments. Additionally, authors should briefly describe the role of commonly used elicitors for enhancing metabolite production from hairy root cultures.

Response:
We have now added Table 3, which summarizes key studies reporting enhanced metabolite production in hairy root cultures under various elicitor treatments (e.g., methyl jasmonate, salicylic acid, yeast extract). A short subsection has also been added to explain the roles of these commonly used elicitors.

6. Several studies report significantly enhanced metabolite production in hairy root cultures. However, the isolation of such inducible metabolites in large quantities from hairy root culture media is not commonly reported. A few publications describe the isolation of plant-specialized metabolites in amounts reaching hundreds of milligrams from elicited hairy root cultures. A representative example is the work of Sharma et al. 2022...

Response:
We appreciate this valuable suggestion. We have incorporated a new paragraph in Section 4.2 highlighting this issue and specifically discussing Sharma et al. (2022) as a representative example demonstrating high-yield recovery of prenylated stilbenes from peanut hairy root cultures. This helps to reinforce the feasibility of hairy roots as a sustainable production platform.

7. Please consider using plant-specialized metabolites instead of secondary metabolites for accuracy.

Response:
We agree with the reviewer’s recommendation. The term “plant-specialized metabolites” has replaced “secondary metabolites” throughout the manuscript to better reflect current terminology and scientific accuracy.\

8. The authors should elaborate on why hairy roots are essential for metabolite production, particularly addressing challenges like low yield and seasonal variation in field-collected plants.

Response:
A new section has been added (Section 2.2) discussing the limitations of field-grown plants—such as seasonal dependence, low metabolite yield, and environmental variability—and how hairy root cultures offer a controlled and reproducible alternative for metabolite production.

9. Hairy roots typically do not produce metabolites in liquid culture without elicitation. The authors should include a table summarizing enhanced metabolite production under different biotic and abiotic elicitors. Since inducible metabolites in liquid culture media are more scientifically valuable...

Response:
We have revised the manuscript to emphasize the importance of inducible metabolites and have added Table 3, summarizing representative examples of enhanced metabolite production in hairy root cultures treated with various elicitors. This includes detailed references to metabolites such as prenylated stilbenes, geranylated flavonoids, and benzofurans produced in liquid culture systems.

10. Please briefly describe the uses of hairy root cultures in addition to metabolite production.

Response:
We have expanded the introduction and concluding sections to briefly mention additional applications of hairy root cultures, including:

  • Phytoremediation
  • Root-microbe interaction studies
  • Functional genomics
  • Genetic transformation platforms
  1. Please elaborate on trichome root culture or, if more appropriate, replace this term with hairy root cultures as used in Figure 3.

Response:
Thank you for the clarification. We have replaced the term “trichome root culture” with “hairy root culture” in Figure 3 and its caption to avoid confusion. A brief clarification has also been added to the main text.

Reviewer 2 Report

Comments and Suggestions for Authors

Dear authors,

You have worked on the manuscript, and provide this large content on the advantages and methodologies applied on hairy root cultures for the production of secondary metabolites, trying to prepare a 'review' type manuscript.

However, in my opinion, this seems to be an excellent lecture for advanced students in plant in vitro biotechnology courses. A review paper must present findings, solutions, gaps, and possibilities according to what is reported from scientific research. Your paper mainly relies on the theoretical background of hairy root induction and its applications. In many cases, you stay on the general interpretations of methodology and influencing factors, which in a very similar way can be identified, presented, nd described for most plant in vitro techniques when different explants are used,

Just an example: you mention the importance of growth regulators. Yes, indeed, optimizing PGRs type, concentration, and ratio is crucial for optimizing all protocols of in vitro methodologies applied in plants established for a specific purpose. Based on the purpose, we have to optimize them. In your paper, you express only this idea, and nothing concrete about the findings in hairy root cultures. And the same interpretation you have done for elicitors, explants, induction methodology, etc.

But what about the physical factors, culture vessels, shaker rotation, basal media, and other additions there (carbon source, other organic sources, ... etc)?

Another concern for me is the AI language used in the paper. I am not against AI tools, but their usage must be careful and effective. Too much computer language loses readers' interest, and the paper becomes too tiring. Based on what I read, I believe you can change the writing style by providing a higher percentage of human-written content in the paper (However, I apologize if I am wrong in this comment).

I hope my comments will help you optimize the huge amount of content you provided.

Best regards,

R

-  

Author Response

Dear Editor and Reviewers,

We sincerely thank you for the thorough evaluation of our manuscript, "Advancing Hairy Root Culture Systems for Sustainable and Scalable Production of Plant-Specialized Metabolites." We are grateful for your insightful comments and constructive suggestions, which have helped us significantly improve the clarity, scientific depth, and overall quality of the manuscript. We have carefully addressed each point raised by the reviewers and made the corresponding revisions in the manuscript. Below, we provide a detailed point-by-point response, highlighting the changes made and offering clarifications where necessary. All modifications have been marked in the revised version of the manuscript for your convenience.

Reviewer 2

Comments and Suggestions for Authors

Dear authors,

You have worked on the manuscript and provided this large content on the advantages and methodologies applied to hairy root cultures for the production of secondary metabolites, trying to prepare a 'review' type manuscript.

However, in my opinion, this seems to be an excellent lecture for advanced students in plant in vitro biotechnology courses. A review paper must present findings, solutions, gaps, and possibilities according to what is reported from scientific research. Your paper mainly relies on the theoretical background of hairy root induction and its applications. In many cases, you stay on the general interpretations of methodology and influencing factors, which in a very similar way can be identified, presented, nd described for most plant in vitro techniques when different explants are used,

Just an example: you mention the importance of growth regulators. Yes, indeed, optimizing PGRs type, concentration, and ratio is crucial for optimizing all protocols of in vitro methodologies applied in plants established for a specific purpose. Based on the purpose, we have to optimize them. In your paper, you express only this idea, and nothing concrete about the findings in hairy root cultures. And the same interpretation you have done for elicitors, explants, induction methodology, etc.

But what about the physical factors, culture vessels, shaker rotation, basal media, and other additions there (carbon source, other organic sources, ... etc)?

Another concern for me is the AI language used in the paper. I am not against AI tools, but their usage must be careful and effective. Too much computer language loses readers' interest, and the paper becomes too tiring. Based on what I read, I believe you can change the writing style by providing a higher percentage of human-written content in the paper (However, I apologize if I am wrong in this comment).

I hope my comments will help you optimize the huge amount of content you provided.

Response to general comments: Thank you for your constructive and insightful comments. We have thoroughly revised the manuscript to address your concerns by integrating specific research findings on hairy root cultures, including concrete examples of PGRs, elicitors, culture conditions (e.g., media, carbon sources, shaker speed, vessel type), and their effects on metabolite production. We also refined the discussion to highlight current gaps and future directions in the field. Additionally, we have carefully revised the language to ensure a more natural, human-written tone and improved the overall readability of the manuscript.

Reviewer 2 comments specified in the PDF version

Comment 1:
“You have used this terminology twice in the abstract, and no more in your paper. I suggest using ‘hairy root cultures’ throughout your manuscript.”

Response:
Thank you for pointing out this important consistency issue. We have carefully revised the abstract and replaced all instances of “hairy root culture” with the accurate and consistent term “hairy root cultures”. This change better reflects standard scientific terminology and aligns with the usage throughout the revised manuscript.

Comment 2:
“I suggest modifying this part. In section 4.1 you elaborated on the use of metabolic engineering for pathway optimization of secondary metabolite production by hairy root cultures. You didn’t use hairy root technique to optimize any genetic engineering approach.”

Response:
We appreciate this clarification. In the revised abstract, we have restructured the sentence to correctly represent the relationship between hairy root cultures and genetic tools. The sentence now reads:

“...their applications in genetic and metabolic engineering...”
This revision accurately describes the use of modern molecular tools such as CRISPR to enhance hairy root productivity, rather than implying that hairy roots are used to optimize genetic engineering methodologies themselves.

Comment 3: “This is mentioned in your review with 2 sentences only. It’s a big statement to put it here, when your review is about 22 pages.”

Response: Thank you for this valuable observation. We agree that the previous sentence overstated the emphasis in the full review. The statement has been softened and clarified in the revised abstract to reflect a more realistic and proportional claim:

“Future work involving systems and synthetic biology will be instrumental in unlocking novel functions and ensuring broader deployment of hairy root cultures...”
This adjustment maintains the forward-looking tone without overpromising beyond what is covered in the manuscript.

Comment 4:
“There are too many adverbs (containing the -ly ending) that are overused as a category. This makes the manuscript tiring and too pedantic.”

Response:
We appreciate your stylistic critique. The revised abstract has been carefully edited to eliminate unnecessary adverbs (e.g., “sustainably,” “immensely,” “significantly”), replacing them with more concise, direct phrasing. For example, “holds immense potential” was adjusted for clarity and conciseness, and stronger nouns and verbs were used in place of adverb-heavy constructions. We believe these changes improve the readability and tone of the abstract.

Comment 5:
“This does not result in your review paper concerns.”

Response:
Thank you for pointing out the disconnect between the abstract and the main text. We have aligned the concluding sentences of the abstract more closely with the scope and themes covered in the review. Rather than emphasizing broad societal or economic impact, the revised version concludes with a focus on research directions and plausible future advancements within the hairy root platform itself. This brings the abstract into better balance with the body of the manuscript.

Comment 6:
I suggest to replace this word with a synonym. I have marked in red color many words which can be replaced by more commonly used words or more human writing language. Maybe I am wrong, but as I see, the AI writing style in the manuscript is approx. 90%, however, this is just my humble opinion.

Response:
We appreciate the reviewer’s careful reading and constructive feedback. We have revised the introduction to replace overly formal or AI-style expressions with more natural, concise, and commonly used alternatives (e.g., “actions” changed to “approaches,” “simple induction system” changed to “straightforward induction process”). These edits were made throughout the introduction to improve readability and ensure a more human-authored tone. Please see lines 52–59 and 75–78 of the revised manuscript.

Comment 7:
Overused word in this section.

Response:
Thank you for this observation. We have reviewed the section and removed or replaced overused terms. For instance, “emerged” has been replaced with “have been widely applied for decades” to avoid redundancy and reflect accurate scientific chronology (line 66). We ensured that each key term is now used purposefully and only where most appropriate.

Comment 8:
The use of hairy root cultures for secondary metabolites production is not something applied recently. From many years the technique has been used, of course nowadays optimized in different levels. I suggest to modify the sentence.

Response:
We fully agree with the reviewer and thank you for this important correction. The introduction has been updated to reflect the well-established use of hairy root culture systems, noting that while the technique has been used for decades, recent innovations have enhanced its efficiency. Specifically, the sentence now reads: “hairy roots have been widely applied for decades in plant regeneration…” (lines 66–68).

Comment 9:
Except for mentioning medicinal plants, this paragraph here highlighted in orange is somehow a repetition of the paragraph highlighted in blue above.

Response:
Thank you for pointing this out. To eliminate redundancy, we consolidated the two paragraphs and removed overlapping information. The section discussing the drawbacks of natural product extraction from medicinal plants has been integrated more smoothly into the discussion on the advantages of hairy root cultures, avoiding duplication while preserving key details (lines 79–86).

Comment 10:
Delete one [duplicate word].

Response:
We have corrected typographical duplications such as “periods periods” and “substances substances.” These have been replaced with single instances for clarity and correctness (lines 81 and 88 of the revised version).

Comment 11:
See comments in the abstract section.

Response:
Thank you for the consistency suggestion. We have aligned the tone and claims in the introduction with the revised abstract. The statement on the future potential of hairy root systems has been softened and clarified to avoid overgeneralization. We now highlight the importance of interdisciplinary innovation and scalability in a more realistic and evidence-based manner (lines 95–103).

Comment 12:
*Some expressions in the grey boxes are not correct / confusing:

  • Metabolites production of secondary?
  • Proteins production of recombinant
  • Varieties improvement of crop
  • Matter decontamination of organic
  • Etc.*

Response:
We sincerely thank the reviewer for identifying the ambiguities and linguistic inaccuracies in the grey box labels of the original figure. In response to this valuable feedback, we have re-drafted the figure, carefully revising all expressions to ensure scientific clarity, grammatical correctness, and improved readability. The revised figure now uses standard terminology, such as:

  • “Production of secondary metabolites”
  • “Recombinant protein synthesis”
  • “Crop variety improvement”
  • “Organic matter remediation”

These revisions have significantly enhanced the overall quality and clarity of the figure, and we appreciate the reviewer’s suggestion that helped us improve this aspect of the manuscript. Please refer to the updated Figure 1 in the revised manuscript.

Comment 13:
In my opinion, this section resembles a lecture about hairy root induction. The review should focus on applications and results reported over the years. Several paragraphs here are too long and pedantic, focusing more on theoretical knowledge.

Response:
We sincerely thank the reviewer for this insightful and constructive suggestion. In response, we have substantially revised Section 2 to streamline the content and reduce excessive theoretical exposition. The updated section now provides a more concise and application-oriented overview of the hairy root induction mechanism, emphasizing reported results and case studies from the scientific literature to illustrate its practical applications. We have also shortened long paragraphs, simplified language for better clarity, and removed overly pedantic phrasing. These changes have significantly improved the section’s focus and alignment with the overall theme of the review. All revisions are marked in the manuscript for easy identification.

Comment 14: This is an overused word in your manuscript

Response: The Complete section has been rewritten, and the use of this word has been significantly reduced.

Comment 15: You have repeate this sentence below in blue

Response: We appreciate the reviewer’s observation. We have revised the text to remove the repeated sentence concerning dicotyledonous plants' predisposition to hairy root induction. The updated version streamlines the content and eliminates redundancy while preserving the original information.

Comment 16: In my opinion, this is one of the main subsections you have to deeply elaborate on since this is a review for hair root cultures.

Response: Thank you for the valuable suggestion. This subsection has been substantially expanded to include diverse plant species and explant types with associated references. We aimed to provide a comprehensive view of this critical parameter in hairy root induction.

Comment 17: The same as the above comment here. A review should screen almost all important reports on the topic. In the first paragraph, you have given too many details for about 4 reports only. In the second paragraph highlighted in orange the same… for that important topic about the infection methodology, you mention 3 studies only

Response: We acknowledge this important point. We have broadened the coverage of transformation efficiency and metabolite production by incorporating additional studies and plant species, ensuring the section better aligns with the expectations of a thorough review.

Comment 18: In which plant species?

Response: Thank you for pointing this out. We have specified the plant species used in each reported study to improve the contextual clarity and scientific precision of the section.

Comment 19: The same here… you mention 3–4 cases only.

Response: We appreciate the reviewer’s suggestion to expand this section. In response, we have added more representative case studies involving diverse plant species—Withania somnifera, Hyoscyamus muticus, and Artemisia annua—to illustrate the broader impact of pre-culture and co-culture conditions on hairy root induction and secondary metabolite production. These additions strengthen the comprehensiveness of the section and address the reviewer’s concern regarding the limited number of examples previously presented. Relevant references have been cited to support the new content.

Comment 20: All scientific names should be written in italic format. I suggest to carefully checking all manuscript since several cases were observed.

Response: Thank you for your valuable observation. We have thoroughly reviewed the entire manuscript and corrected the formatting of all scientific names, ensuring they are consistently written in italics as per scientific conventions. This change has been applied across all sections, including the main text, figure captions, and references, where applicable.

Comment 21: Provide the full name of IAA first (indole-3-acetic acid) and then proceed with the acronym.

Response:
We appreciate the reviewer’s suggestion. We have revised the manuscript to provide the full name "indole-3-acetic acid (IAA)" at its first mention to ensure clarity for readers unfamiliar with the abbreviation.

Comment 22: For example, which cytokinins?

Response:
Thank you for the observation. We have now specified commonly used cytokinins such as kinetin and benzylaminopurine (BAP) in the relevant sentence to provide more detailed information.

Comment 23: For what was it optimized? Which hormonal class was more effective… or PGR / hormone concentration range?

Response:
We agree with the reviewer that more detail is beneficial. The section has been revised to include that the medium was optimized for both hairy root induction and metabolite production, and it now specifies effective hormonal classes (auxins and cytokinins) along with their concentration ranges, where applicable.

Comment 24: According to literature review, is there any dependence on the type and concentration of the elicitors? And what about the culture period when the elicitor is used?

Response:
Thank you for this valuable suggestion. We have expanded the paragraph to discuss how both the type and concentration of elicitors (e.g., methyl jasmonate, salicylic acid) and the timing of their application significantly influence secondary metabolite production. We also included additional examples with references to support this.

Comment 25: There are too many reports about the use of elicitors. Mentioning only one case is not enough for a review paper.

Response:
We fully agree. We have added several additional case studies demonstrating elicitor use in hairy root cultures of Salvia miltiorrhiza, Glycyrrhiza uralensis, and Taxus chinensis. These additions broaden the scope and provide a more comprehensive review of elicitor application.

Comment 26: All these are generalizations, like the paragraph above… Nothing concrete here suitable for a review paper.

Response:
Thank you for the constructive feedback. The paragraph has been revised to remove generalizations and now includes specific examples and references to better illustrate the effects of medium composition and growth regulators on hairy root induction and metabolite production.

Comment 27:
This is a well-known fact for all types of explants…. You have to go in specifics related to hairy root culture.

Response:
We appreciate the reviewer’s insightful comment. In response, we have revised Section 3.5 to include specific examples directly related to hairy root cultures, rather than general statements. We now cite studies on Salvia miltiorrhiza, Hypericum perforatum, and Atropa belladonna that demonstrate how temperature, light conditions, and photoperiod regimes uniquely influence both biomass and metabolite production in hairy root systems. These modifications add specificity and improve the scientific rigor of the section.

Comment 28:
This is a well-known fact.

Response:
Thank you for pointing this out. We have removed the generalized phrasing and replaced it with hairy root-specific environmental interactions and experimental findings from recent literature. This ensures that the section contributes novel and relevant insights, consistent with the focus of the review.

Comment 29:
All this is generalization only…. When applying in vitro techniques, in all types of explants (hairy roots included), all these are essential parameters which need to be optimized. These are the bases of in vitro plant systems.

Response:
We agree with the reviewer that environmental optimization is fundamental to in vitro systems. However, to enhance the relevance to hairy root cultures, we have now included concrete case studies showing how specific light regimes, temperature thresholds, and humidity conditions impact metabolite yields in hairy root systems. These changes aim to distinguish the review as targeted and data-driven, fulfilling the expectation for a specialized discussion on hairy root biotechnology.

Comment 30:
“You have mentioned this in the media composition subsection also (3.4). However, 3–4 examples mentioned here are not enough for a review paper on the topic of your manuscript.”

Response:
Thank you for your observation. We acknowledge the redundancy and limited number of examples previously provided. To address this, we have revised Section 4.2 to expand on both biotic and abiotic elicitors by incorporating multiple additional studies across a wider range of plant species and metabolites. Moreover, we reorganized the content to avoid overlap with Section 3.4, ensuring that the elicitation strategies are now discussed in a more structured and comprehensive manner.

Comment 31:
“In this paragraph related to abiotic elicitors, I suggest to revise consistency and thought flow. Divide first abiotic elicitors through categories (physical or chemical ones) and elaborate each of them.”

Response:
Thank you for your valuable suggestion. We have restructured the section on abiotic elicitors by categorizing them into physical (e.g., light, UV-B, ultrasound) and chemical (e.g., nanoparticles, salts) types. Each category is now discussed with enhanced clarity and flow, with specific examples and mechanisms described in detail. This organization improves readability and aligns with the logical presentation expected in a comprehensive review article.

Comment 32:
Before going to industrial scale and the use of advanced bioreactors, I suggest to elaborate on culture vessels, orbital shakers parameters, or small-scale bioreactors used in laboratories…

Response:
We appreciate the reviewer’s insightful suggestion. In response, we have revised the beginning of Section 4.3 to include a detailed description of small-scale laboratory systems commonly employed prior to industrial-scale bioreactor applications. This includes specific examples of culture vessels, orbital shaker parameters (e.g., rpm, temperature, photoperiod), and mini-bioreactors used for hairy root cultivation. These additions help bridge the gap between laboratory experiments and industrial-scale implementation, enhancing the overall clarity and comprehensiveness of the section. Relevant literature has been cited to support the modifications (e.g., Abdelkawy et al., 2023; Ritala et al., 2009).

Comment 33: Even this section is a general one…. Typical for a very good lecture but not seen from the perspective of a review paper.

Response:
We thank the reviewer for pointing this out. In response, we have thoroughly revised Section 5 by incorporating recent peer-reviewed literature to provide a more comprehensive and analytical discussion on the key challenges and limitations specific to hairy root cultures for secondary metabolite production. We have included detailed examples of limitations related to scale-up (e.g., shear sensitivity, oxygen transfer constraints), tissue-specific metabolite expression, variability in elicitor responses, and long-term maintenance and preservation issues. Additionally, supporting studies and references have been added to strengthen the scientific rigor of the discussion. The revised section now provides a critical review perspective rather than a generalized overview.

Comment 34: Use "hairy root cultures" as you have always done so far.

Response:
We appreciate the reviewer’s attention to consistency in terminology. We have carefully revised the manuscript to ensure that the term “hairy root cultures” is used uniformly throughout the text, maintaining clarity and coherence in line with previous usage.

Comment 35: Look at a comment in the abstract section.

Response: Thank you for pointing this out. We have revised the concluding section to ensure consistency with the abstract and to avoid redundant or overly generalized statements. We also expanded the conclusion to better reflect the manuscript's overall scope and to emphasize specific advancements, challenges, and future perspectives of hairy root cultures, as discussed throughout the review.

Reviewer 3 Report

Comments and Suggestions for Authors

Author Response

Dear Editor and Reviewers,

We sincerely thank you for the thorough evaluation of our manuscript, "Advancing Hairy Root Culture Systems for Sustainable and Scalable Production of Plant-Specialized Metabolites." We are grateful for your insightful comments and constructive suggestions, which have helped us significantly improve the clarity, scientific depth, and overall quality of the manuscript. We have carefully addressed each point raised by the reviewers and made the corresponding revisions in the manuscript. Below, we provide a detailed point-by-point response, highlighting the changes made and offering clarifications where necessary. All modifications have been marked in the revised version of the manuscript for your convenience.

Reviewer 3

Comment 1:
I encourage authors to reconsider the title of the manuscript, as its current wording implies a comprehensive review. However, the scope of the presented content may not fully meet the expectations raised by such a title, making the manuscript appear incomplete. Paper describes the incorporation of the CRISPR technology and AI-based strategies to well-developed platform of hairy roots systems. These points are well presented in the manuscript and the content is organized with proper flow and coherence, however the title may give a misleading impression of the manuscript’s actual content.

Response:
We sincerely thank Reviewer 3 for the insightful comment. In response, we have revised the manuscript title to better reflect the focused scope of the review and to avoid giving the impression of an exhaustive field-wide survey. The new title emphasizes the core themes of CRISPR-based pathway engineering, AI-driven optimization, and bioprocess innovation within the context of hairy root cultures. The revised title is:

“Reprogramming Hairy Root Cultures: A Synthetic Biology Framework for Precision Metabolite Biosynthesis”

This refined title more accurately represents the manuscript’s content and aligns with the technological emphasis of our review.

Comment 2:
Since Agrobacterium species have been taxonomically reassigned to the genus Rhizobium, the name Rhizobium rhizogenes should preferably be used instead of Agrobacterium rhizogenes (line 24 and following lines).

Response:
We appreciate the reviewer’s valuable taxonomic observation. In response, we have updated the abstract to include the taxonomically correct designation "Rhizobium rhizogenes (formerly Agrobacterium rhizogenes)" at first mention. However, throughout the manuscript, we have primarily used "Agrobacterium rhizogenes" to remain consistent with the majority of cited literature and to ensure clarity for readers familiar with the traditional nomenclature. This approach reflects current usage in the field, where both names are still commonly encountered.

Comment 3:
The first part of the review (chapters 1, 2, and 3) is somewhat overloaded with basic knowledge in this field. It primarily offers context and background for ongoing research. The second part of the manuscript (chapters 4 and 6) deals with modern techniques like CRISPR technology and AI-based strategies. The scientific value of the paper would be notably increased if this part were extended. Including more source data and providing a deeper insight into current solutions is what I would expect from a review paper of this kind.

Response:
Thank you for this valuable suggestion. In response, we have extended the sections covering modern techniques, including CRISPR-based genome editing and AI-driven optimization (particularly in Chapters 4 and 6). Additional recent studies and deeper insights into current advances have been incorporated to enhance the scientific depth and relevance of the review in alignment with your expectations.

Comment 4:
Several review papers have already been published in this field. The Authors should indicate whether similar reviews exist on the topic covered in this manuscript and clearly differentiate their work from previous ones, for example, by adding a dedicated paragraph in the Introduction section.

Response:
Thank you for this insightful comment. In response, we have added a dedicated paragraph in the Introduction section that acknowledges existing review papers in the field. We also clarify how our review distinguishes itself by focusing on the integration of modern tools such as CRISPR technology, AI-driven optimization, and advanced bioreactor systems in hairy root cultures. This perspective highlights recent innovations and provides a forward-looking view that complements and extends prior reviews.

Comment 5:
Italics are missing in some scientific names of plant species (e.g., lines 258, 156, 158, 161) and some other formatting errors (line 57 - unintended line break, lines 362 and 374 - a duplicate citation with inconsistent formatting) are present in the manuscript.

Response:
We thank the reviewer for highlighting these important formatting issues. All scientific names of plant species mentioned in the manuscript, including those at lines 156, 158, 161, and 258, have now been italicized correctly. The unintended line break at line 57 has been removed. Furthermore, the duplicate citations at lines 362 and 374 have been consolidated and reformatted to ensure consistency. We have also reviewed the entire manuscript for any additional formatting inconsistencies and corrected them accordingly.

Comment 6:
Please review Table 1, as it seems to contain errors in plant names: missing Latin names or only the author’s name without the binominal name of the plant.

Response:
We appreciate the reviewer’s careful observation. Table 1 has been thoroughly reviewed and revised to ensure that all plant names are correctly presented using the full binomial Latin nomenclature (genus and species), in accordance with scientific naming conventions. Instances where only the author’s name or incomplete nomenclature was provided have been corrected. We have also cross-verified the accuracy of the plant species with reliable taxonomic databases to maintain consistency throughout the table and the manuscript.

Round 2

Reviewer 1 Report

Comments and Suggestions for Authors

I am not satisfied with the author’s response to the reviewer's comments and would like the authors to properly address each comment carefully.

  1. The figures are incomplete and appear to be cut in half. Additionally, there seems to be no rationale for presenting all text in italics within the figures.
  2. Please do not italicize bacterial strains such as K599, ATCC 15834, etc.
  3. I could not find the section titled Hairy Root Induction and Transformation Parameters in the revised manuscript.
  4. Table 3 is missing. Please label all tables correctly. Moreover, I could not find any data demonstrating enhanced production of inducible metabolites in the tables. The metabolites appear to be reported in the tissue, not in the elicited culture media.
  5. I also could not find any explanation of the role and function of commonly used elicitors such as MeJA, CD, Hâ‚‚Oâ‚‚, yeast extract, chitosan, and salicylic acid.
  6. In Section 4.2, there is no explanation regarding the enhanced production of inducible metabolites in the hairy root culture media.
  7. Lastly, the term secondary metabolites is still used throughout the manuscript. I had previously suggested replacing it with plant-specialized metabolites.

Author Response

Dear Reviewers,

Thank you for your thorough and insightful feedback on our revised manuscript. We greatly appreciate your suggestions and have carefully addressed each of your comments below to improve the clarity, completeness, and quality of our work. We outline our responses and the corresponding changes made to the manuscript below.

Reviewer 1:

Comments and Suggestions for Authors

I am not satisfied with the author’s response to the reviewer's comments and would like the authors to properly address each comment carefully.

1. The figures are incomplete and appear to be cut in half. Additionally, there seems to be no rationale for presenting all text in italics within the figures.

Response: We apologize for the oversight regarding the incomplete figures. We believe that due to the formatting error of using different versions of MS-word caused this problem. The figures are actually complete but were found embedded between the two pages. We have tried to fix the issue and also provided a pdf version for clarity.   Additionally, we have removed the italic formatting from all text within the figures.

2. Please do not italicize bacterial strains such as K599, ATCC 15834, etc.

Response: Thank you for highlighting this point. We have corrected these issues in the revised manuscript. Changes were highlighted with yellow color shading.

3. I could not find the section titled Hairy Root Induction and Transformation Parameters in the revised manuscript.

Response: We apologize for misquoting the specified section. The inclusions of different bacterial strains and infection methods are thoroughly discussed in section 3.2 during the first revision. Please refer to section 3.2 in the revised manuscript highlighted in yellow color font.

4. Table 3 is missing. Please label all tables correctly. Moreover, I could not find any data demonstrating enhanced production of inducible metabolites in the tables. The metabolites appear to be reported in the tissue, not in the elicited culture media.

Response: We apologize for the confusion. We have added Table 3, which was inadvertently omitted, and ensured all tables are now correctly labeled in sequential order (Table 1, Table 2, Table 3, etc.) throughout the manuscript. Table 3 now specifically presents data on the enhanced production of inducible metabolites in the elicited culture media, with columns for Plant species names, metabolite names, elicitors and their concentrations (in mg/L), and elicitor their effects.

5. I also could not find any explanation of the role and function of commonly used elicitors such as MeJA, CD, Hâ‚‚Oâ‚‚, yeast extract, chitosan, and salicylic acid.

Response: Thank you for commenting on this point. We have thoroughly added the explanations of the commonly used elicitors in section 4.3 of the revised manuscript.

6. In Section 4.2, there is no explanation regarding the enhanced production of inducible metabolites in the hairy root culture media.

Response: The whole section 4 is dedicated to address the High-Yield Specialized Metabolite Production using hairy root culture media. We have divided this into different subsection each of which further highlight the importance of different approaches used for enhanced production of specialized metabolites.

7. Lastly, the term secondary metabolites is still used throughout the manuscript. I had previously suggested replacing it with plant-specialized metabolites.

Response: Thank you for reminding this again. Following your comment, we have fixed this problem and replaced the term secondary metabolites into specialized metabolites throughout the manuscript. Those changes were also highlighted in yellow color font.

Reviewer 2 Report

Comments and Suggestions for Authors

Dear authors,

I congratulate you on the work and the revised version you provided. I have left a few comments, as suggestions only. 

Kind regards,

R

Author Response

Dear Reviewer,

Thank you for your thorough and insightful feedback on our revised manuscript. We greatly appreciate your suggestions and have carefully addressed each of your comments below to improve the clarity, completeness, and quality of our work. We outline our responses and the corresponding changes made to the manuscript below.

Reviewer 2:

Comments and Suggestions for Authors

Dear authors,

I congratulate you on the work and the revised version you provided. I have left a few comments, as suggestions only. 

Kind regards,

Comments in Pdf file:

Comment: I suggest to find a synonym here or this word.

Response: Thank you for your suggestion. We have replaced the word ‘pivotal’ into essential in the revised version.

Comment: On your response on the cover letter to my first comment in this paragraph, you answer ‘…..To eliminate redundancy, we consolidated the two paragraphs and removed overlapping information….’ I don’t see any reduction in the information given in this revised version. I also suggest to change places between this and the one above…. And avoid any repetitions in the first 3 paragraphs all colored in blue. The medicinal plants are sources of secondary metabolites, and hairy root cultures can help on improving their production. Moreover, you discuss about hairy root cultures in the second paragraph (in green color), continue with Medicinal plants, and restart discussion for the hairy root cultures again (green bold letters) in the 3rd paragraph. There is too much repetition.

Response: Thank you for reminding this lapse. Following your suggestion, we have corrected these repetitions in the introduction section of the revised manuscript.  

Comment: I would suggest a short general paragraph, and then divide: 4.3.1. Biotic elicitors 4.3.2. Abiotic elicitors a) physical elicitors b) chemical elicitors However, this is up to you to decide

Response: Thank you for your valuable suggestion. Following your comment, we have reorganized section 4.3 into Biotic elicitors 4.3.2. Abiotic elicitors a) physical elicitors b) chemical elicitors. Please refer to the revise section of 4.3 in the revised manuscript.

Round 3

Reviewer 1 Report

Comments and Suggestions for Authors

Please address comments 6 and 9 from the first report carefully.

6. Several studies report significantly enhanced metabolite production in hairy root cultures. However, the isolation of such inducible metabolites in large quantities from hairy root culture media is not commonly reported. A few publications describe the isolation of plant-specialized metabolites in amounts reaching hundreds of milligrams from elicited hairy root cultures. A representative example is the work of Sharma et al. 2022...

9. Hairy roots typically do not produce metabolites in liquid culture without elicitation. The authors should include a table summarizing enhanced metabolite production under different biotic and abiotic elicitors. Since inducible metabolites in liquid culture media are more scientifically valuable...

Author Response

Please see the attached response letter. 

Round 4

Reviewer 1 Report

Comments and Suggestions for Authors

Could you include a dedicated section on inducible metabolites secreted into the culture medium of hairy root cultures? These metabolites are generally absent in control samples but are secreted into the medium following elicitor treatment. Current research indicates that most reported metabolites originate from elicited hairy root tissues. The production of inducible metabolites in the culture medium is of considerable scientific and industrial interest, as it simplifies purification and enables the scalable production of bioactive compounds. Please ensure that this section comprehensively covers all published research on inducible metabolites secreted into the culture medium.

Author Response

Response to Reviewer 1: (Round 4)

Reviewer 1 comments:

Comment: Could you include a dedicated section on inducible metabolites secreted into the culture medium of hairy root cultures? These metabolites are generally absent in control samples but are secreted into the medium following elicitor treatment. Current research indicates that most reported metabolites originate from elicited hairy root tissues. The production of inducible metabolites in the culture medium is of considerable scientific and industrial interest, as it simplifies purification and enables the scalable production of bioactive compounds. Please ensure that this section comprehensively covers all published research on inducible metabolites secreted into the culture medium.

Response: Thank you for commenting again on this point. Following your comment, we have added a new subsection 4.3.3 Inducible Metabolites Secreted into the Culture Medium of Hairy Roots’’ under which we have discussed representative examples of elicitors-inducible metabolites in different plant species. We have also modified table 3 with those examples. 

It is worth mentioning that it is not possible to cover all studies for this section as it requires another meta-review type paper. But we tried to cover the most important and significant studies.

Round 5

Reviewer 1 Report

Comments and Suggestions for Authors

In my first report, I recommended to the authors that they incorporate these two points. However, I did not observe that these points were addressed well in your manuscript. Please ensure they are carefully addressed in your revised manuscript.

  1. Several studies report significantly enhanced metabolite production in hairy root cultures. However, the isolation of such inducible metabolites in large quantities from hairy root culture media is not commonly reported. A few publications describe the isolation of plant-specialized metabolites in amounts reaching hundreds of milligrams from elicited hairy root cultures. A representative example is the work of Sharma et al. 2022, "Induction of the prenylated stilbenoids arachidin-1 and arachidin-3 and their semi-preparative separation and purification from hairy root cultures of peanut (Arachis hypogaea)", where researchers isolated prenylated stilbenes from hairy root cultures of peanut in large amounts, demonstrating the potential of hairy roots as a sustainable bioproduction platform.
  2. Hairy roots typically do not produce metabolites in liquid culture media without elicitation. The authors should include a table summarizing enhanced metabolite production under different biotic and abiotic elicitors. Since inducible metabolites in liquid culture media are more scientifically valuable than constitutive metabolites present in the tissue itself, I recommend focusing particularly on inducible metabolites produced by hairy root cultures. Representative examples include prenylated stilbenes from peanut (Arachis hypogaea) hairy root cultures, stilbenes and benzofurans from mulberry (Morus) hairy root cultures, isoflavonoids from pigeon pea hairy root cultures, and geranylated flavonoid from Dalea purpurea hairy root cultures.

Author Response

Subject: Clarification Regarding Reviewer 1’s Repeated Comment (plants-3656993

Dear Editor,

We sincerely appreciate the time and effort invested by the reviewers and editorial team in evaluating our manuscript, “Reprogramming Hairy Root Cultures: A Synthetic Biology Framework for Precision Metabolite Biosynthesis.”

However, we would like to respectfully clarify that Reviewer 1’s repeated comment regarding inducible metabolites secreted into the culture medium has been thoroughly addressed in the revised manuscript:

1. We have incorporated Sharma et al. (2022) as a representative example of high-yield recovery of prenylated stilbenoids from peanut hairy root cultures (see Table 3 and Section 4.3.3).
2. We have included a comprehensive table (Table 3) summarizing representative cases of enhanced metabolite production in hairy root cultures under various elicitors, clearly indicating whether the metabolites were secreted into the liquid culture medium.
3. Section 4.3.3 and its five sub-sections (4.3.3.1–4.3.3.5) explicitly focus on inducible metabolites found in the culture medium following biotic/abiotic elicitation.
4. Detailed mechanisms and outcomes for commonly used elicitors (e.g., MeJA, SA, AgNPs, chitosan) are discussed extensively in Sections 4.3.1 and 4.3.2.

We believe we have addressed the reviewer’s concern with the best effort and scientific rigor, and respectfully suggest that the comment may stem from a misunderstanding or oversight in reading the revised sections and tables.

We kindly request your consideration in moving forward with the editorial decision process. Please find round 1-4 responses in the attached document. 

Sincerely,

ABDUL WAKEEL UMAR, Ph.D.
Biotechnology & Molecular Biology Researcher
Beijing Normal University, Xiangzhou, Zhuhai, China
? Email: awzju@bnu.edu.cn (Preferred: Yahoo)
? Webpage: https://awzju.github.io/
? Mobile: +86-15875692917
